There are amendments to this paper

# Zebra finches identify individuals using vocal signatures unique to each call type

Julie E. Elie[1] & Frédéric E. Theunissen [1,2]

Individual recognition is critical in social animal communication, but it has not been demonstrated for a complete vocal repertoire. Deciphering the nature of individual signatures across call types is necessary to understand how animals solve the problem of combining, in the same signal, information about identity and behavioral state. We show that distinct signatures differentiate zebra finch individuals for each call type. The distinctiveness of these signatures varies: contact calls bear strong individual signatures while calls used during aggressive encounters are less individualized. We propose that the costly solution of using multiple signatures evolved because of the limitations of the passive filtering properties of the birds' vocal organ for generating sufficiently individualized features. Thus, individual recognition requires the memorization of multiple signatures for the entire repertoire of conspecifics of interests. We show that zebra finches excel at these tasks.

[1] Helen Wills Neuroscience Institute, UC Berkeley, Berkeley, CA 94720, USA. [2] Department of Psychology, UC Berkeley, Berkeley, CA 94720, USA. Correspondence and requests for materials should be addressed to J.E.E. (email: julie.elie@berkeley.edu) or to F.E.T. (email: theunissen@berkeley.edu)

Humans share the ability to recognize familiar individuals using vocal-cues with many other animal species that rely on individual-specific relationships, such as mated pair bonds, the mother−young bond, or the tolerance relationship developed by neighbor territorial animals (e.g. Birds[1–3]; Mammals[4–7]; Amphibians[8,9]; reviewed in refs. [10–12]). Social animals also have complex vocal repertoires comprised of multiple call types that they use to communicate different behavioral states (e.g. alarm, hunger, need for contact). Investigation of individual vocal recognition in animals, including humans, has demonstrated individual discrimination either in single-call types (e.g. ref. [13].) or in a few call types[7,14–21] but yet, it is unknown whether individual recognition is present for an entire vocal repertoire or, for humans, irrespective of the vocalization produced may it be laughing, speaking, shouting or whispering[22]. If vocal recognition was to exist throughout a repertoire, it would provide a unique opportunity to study the nature of the individual signature given the particular physical constrains of sound production and the ecological pressure to also produce distinctive multiple call types.

The percept of the human voice results in part from the combination of fundamental frequency modulations determined by the properties of the vocal source, and of specific spectral shaping by the vocal tract[12,23]. Individual information in animals could similarly be the result of morphological individual differences in the vocal apparatus that would cause similar passive shaping of the sound across all calls in one's repertoire: the passive-voice-cues[24]. In support of that hypothesis, vocal tract resonances have been found to provide reliable cues to an individual's identity in some calls of nonhuman mammals[14,15,24,25]. Alternatively, or, in addition to passive-voice-cues, humans and animals can actively control their vocal organ not only to produce the different utterances or call types that carry different meanings[26], but also to either exaggerate their voice features, active-voice-cues (red deer[27], humans[28]), or to implicitly advertise one's identity, signature cues (Dolphins' whistles[5,29], and humans' stereotypical volitional laughing bouts[30]). The nonexclusive passive-voice, active-voice, and signatures strategies have yet to be explored when studying individual recognition across a repertoire. Here, we will contrast the use of passive-voice-cues (called simply voice from now on) to signature cues.

The zebra finch is a highly vocal social songbird that relies on a set of 11 call types to communicate different behavioral states, intents or needs[31,32]. Because the coding of identity has been shown independently for some of these call types (Begging calls[33,34]; Distance calls[1,32,35–41]; Song[42–45] and soft calls[46,47] but see ref. [48] for Long Tonal calls), it is a good model to investigate mechanisms of identity coding through a whole repertoire. Here, using operant conditioning design to measure behavioral discriminability and acoustic analyses to determine the features that carry individual information, we perform a thorough comparison of individual signatures across the repertoire of the zebra finch. We find that zebra finches do not produce signatures that generalize across call types but instead generate call-type-specific individual signatures. The strength of this signature varies across call types and is stronger for social contact calls. Our behavioral experiments show that birds are able to discriminate the identity of vocalizers for all calls of the adult repertoires and for the two call types unique to juveniles. Birds are also able to identify a vocalizer irrespective of call type, a task that requires the memorization of a set of vocal signatures.

## Results

**Experimental design.** We used a modified go−no go task to test whether birds can discriminate the identity of vocalizers. In this task, birds maximize a food reward by interrupting the playback

of vocalizations from a nonrewarded (NoRe) vocalizer and refraining to interrupt the playback of vocalizations from a rewarded (Re) vocalizer (Fig. 1). Re and NoRe vocalizations can correspond to two different stimuli (shaping: e.g. a single song of bird A vs. single song of bird B), two sets of renditions of one call type for two different birds (single-call-type test: e.g. randomly chosen renditions of Distance Calls from bird A vs. randomly chosen renditions of Distance Calls from bird B), two sets of renditions from all call types mixed together (all-call-type tests: random renditions of calls from the repertoire of bird A vs. random renditions of calls from the repertoire of bird B). These learning and discrimination tasks are progressively more complex and subject birds participated in all three in sequence (Fig. 2). We tested the discrimination for male vocalizers, female vocalizers, and juvenile vocalizers in randomized order. Finally, the all-call-type tests were performed with calls from vocalizers that subjects had already discriminated in single-call-type tests as well as with calls from unfamiliar vocalizers.

In parallel, we performed bioacoustical analyses and supervised classification to assess the discriminability of vocalizers (see Methods) based on the information present in the acoustic signal as it has been done in other species (e.g. refs. [15,49].). More specifically, 18 predefined acoustic features (PAF) along with the spectrograms were used to characterize the acoustic properties of vocalizations and three types of regularized classifiers were used to investigate the discriminability of ID in vocalizations: linear discriminant analysis (LDA), quadratic discriminant analysis (QDA) and random forest (RF). These analyses were performed for single-call-types as well as for all-call-types grouped together.

**Vocalizer discrimination for each call type.** Figure 3 illustrates the individual signature we found in two call types: the Wsst call (Ws), used in aggressive behaviors, and the Distance call (DC), used to establish contact at distance. Acoustic features become identity cues when they vary between vocalizations emitted by different individuals but are consistent between renditions emitted by the same individual. The spectrograms depict the contrast of rendition-consistency and vocalizer-idiosyncrasy between call types: acoustic features are more individualized in DC than in Ws calls. Despite the lack of apparent identity coding in Ws calls, the performance of a classifier demonstrates that vocalizer discrimination is above chance level for both call types and that DC are highly individualized. The behavioral performance of a zebra finch demonstrates the perception of these individual signatures in both call types and confirms their difference in strength: vocalizer discrimination is significantly above chance for both but the behavioral performance is four times higher for DC than for Ws calls (Fig. 3).

We further investigated vocalizer discrimination for each of the call types in the zebra finch's repertoire. A multiple pair-wise discrimination approach was used for the classifier so that its performance could be compared between call types for which we had a variable number of vocalizers, as well as to behavioral performance measured in the operant task. For each of the call types, a large and significant majority of the pairs of vocalizers tested were discriminated above chance level (Fig. 4a). Classifiers using other feature spaces and nonlinear or noncontinuous decision boundaries yielded similar performance (Supplementary Fig. 1). Thus, an acoustical individual signature is present in all call types and the operant task shows that zebra finches can use this information to discriminate vocalizers irrespective of the call type (Fig. 4b and Supplementary Table 1).

The strength of individual signatures depended on call types. Classifier and behavioral performances were better for vocalizations used to establish contact (DC, LT, and Te; see Fig. 4 for call

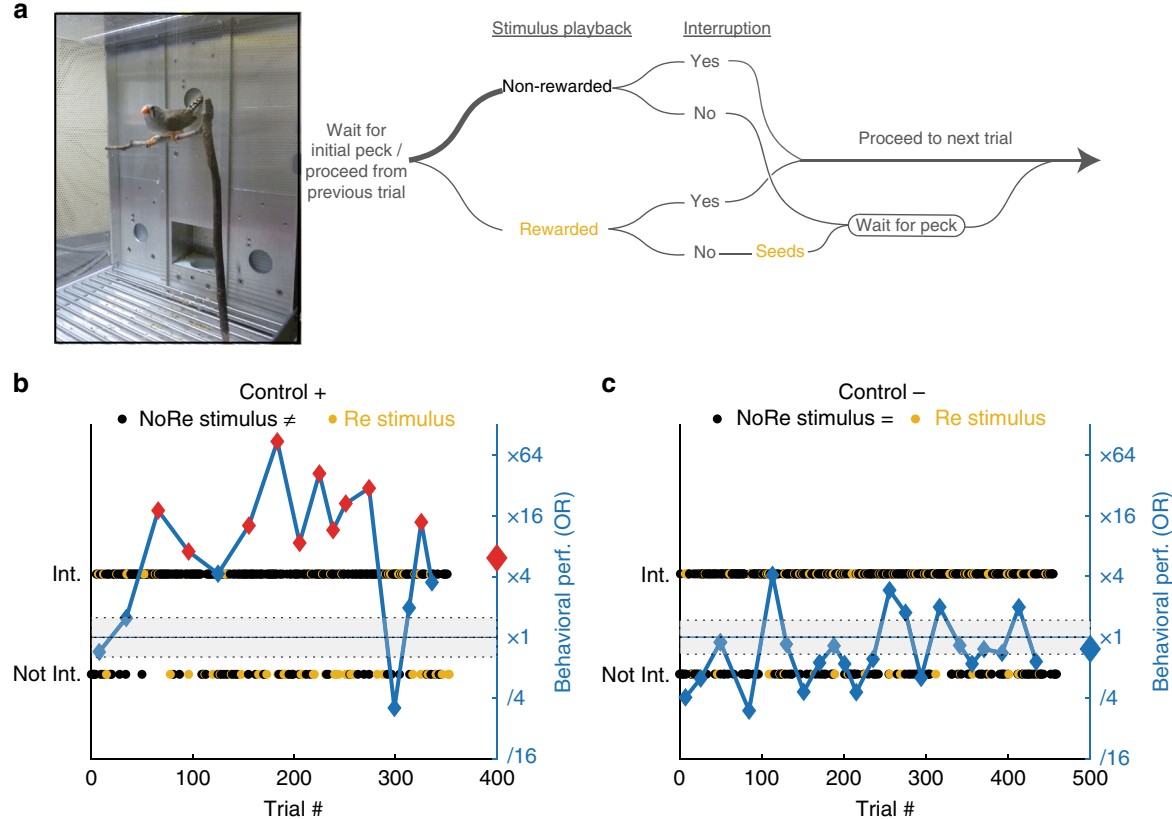

**Fig. 1** Operant conditioning task used for quantifying bird discrimination of vocalizer identity. **a** Bird testing chamber and sequence of events in one trial. Birds learn to peck a keypad to initiate the playback of 6 s sequences of vocalizations from one of two vocalizers. One of the vocalizer is associated with a reward (Re vocalizer) while the other vocalizer is not (NoRe). Twenty percent of trials trigger vocalizations from the Re vocalizer and 80% trigger vocalizations from the NoRe vocalizer. At any time, even during playback, a peck on the keypad triggers a new trial. Interrupting an Re stimulus systematically cancel the access to the reward. Birds maximize their food access by interrupting the NoRe vocalizer and by refraining from interrupting the Re vocalizer (Supplementary Movie 1). **b** Schematic of a subject performing the task. The bird quickly identifies the vocalizers and, accordingly, interrupts more the NoRe vocalizer stimuli (black dots aligned with interrupted (Int.)) than the Re vocalizer stimuli (yellow dots aligned with not-Int.)). The behavioral performance is measured by the odds ratio (OR) of interrupting the NoRe over the Re. OR is calculated at successive time windows (blue line) or over the entire test (large diamond on right $y$-axis). The right $y$-axis uses a logarithmic scale: 'x2' means NoRe vocalizations are twice more likely to be interrupted than Re vocalizations. Red diamonds indicate an OR value significantly different from 1. The gray area depicts the 5−95% quantiles for a random odds ratio for all trials. While locally the OR might not be significant (little blue diamond) although it is above the confidence interval for random interruption, these points contribute to the significance over the entire test (big red diamond). **c** Schematic of a subject in the control condition. This targeted interruption behavior is not observed when the two sets of vocalizations are identical and the reward is randomly assigned to the two vocalizers: OR is randomly varying around ×1

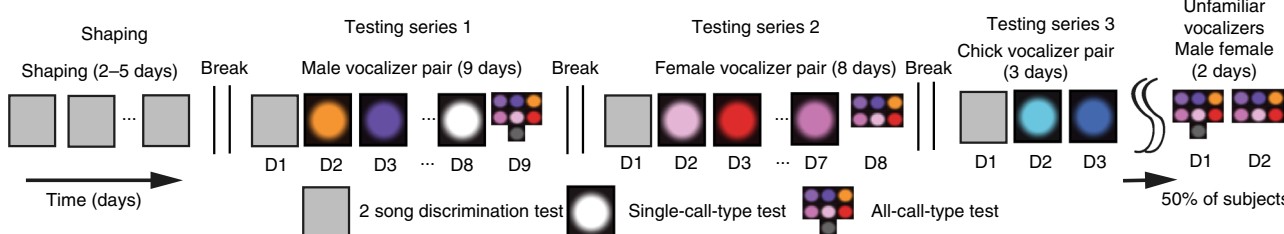

**Fig. 2** Typical series of tasks performed across days by single bird subjects. Each square represents a day of testing with the same set of vocalizations. All tests involve the discrimination of a pair of vocalizers and tests were grouped into three series where either two male vocalizers, two female vocalizers or two chick vocalizers were discriminated. The order of the series was randomized. Gray rectangle corresponds to shaping days where birds were trained and tested with two songs from two different males not used in the testing set. Large single-color circles correspond to single-call-type tests: individual discrimination based on multiple renditions of a single-call type (including song for males, gray circle) is tested on each day. The color codes call types. Multiple small circles correspond to all-call-type tests: individual discrimination based on multiple renditions of all call types was tested on each day. Within a series, the order of consecutive tests was also randomized except for the shaping that was always performed first. At the end of the three series, half of the subjects performed all-call-type tests with new pairs of unfamiliar vocalizers. Single-call-type tests, Familiar all-call-type tests obtained in two of the first three series and Unfamiliar all-call-type tests were analyzed separately

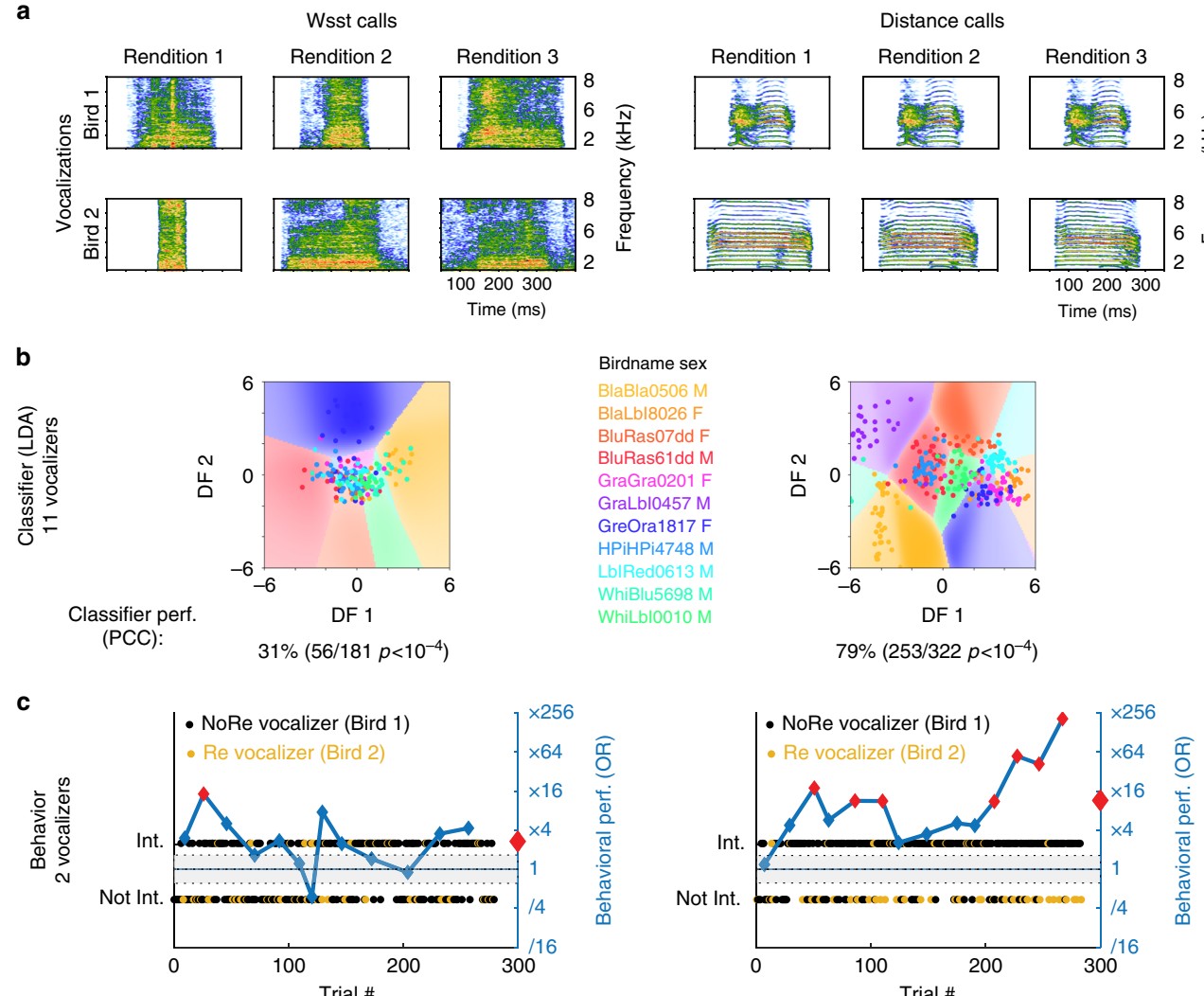

**Fig. 3** Vocalizer discrimination for two call types. Vocalizations (**a**): Spectrograms of aggressive calls (Ws, Supplementary Audio 1) and contact Distance calls (DC, Supplementary Audio 2) from two different males. Regularized linear discriminant analysis (LDA, **b**): LDA was performed on acoustic features to classify calls (each dot corresponds to a call rendition) along the same 11 vocalizers for Ws (*n* = 181) and DC (*n* = 322). Cross-validated classifier performance is shown as Percent of Correct Classification (PCC; chance level, 9.1%). Behavior (**c**): The behavioral performance of one subject for discriminating Bird1 vs. Bird2 is shown for Ws (35 and 31 renditions) and DC (29 and 33 renditions). Each dot corresponds to a stimulus (i.e. a sequence of calls). The behavioral performance as measured by the odds ratio (OR, see Fig. 1) is given by the blue line. Red diamonds indicate an OR value significantly different from 1. The gray area depicts the 5−95% quantiles for a random odds ratio for all trials

labels) and for categories used to sound the alarm (Tu and Th) than vocalizations communicating aggressive intents (Ws calls; Fig. 4a, b). The strength of individual signature can be particularly impressive for contact calls: with DC, the classifier reaches 100% correct classification on cross-validation data (testing set: *n* = 3514 calls) for 64/171 pairs of vocalizers, and the average classification of these 171 vocalizer pairs was 95%. Thus, birds produce strong individual signatures in contact calls which could ensure an acoustic network is maintained irrespective of the distance between individuals and of the access to visual cues[40,41]. This strong individual signature also allows for the recognition of multiple individuals (as illustrated by the Discriminant Analysis shown in Fig. 3b). Therefore, in addition to identifying their mate, zebra finches could use Distance call to identify multiple individuals. This multiplicity of individual recognition remains to be demonstrated in a natural context[11].

We also investigated if the difference in behavioral performance between call types could be explained by differences in the acoustical signature as revealed by the classifier. Figure 4c plots

the behavioral performance of subjects against the classifier performance for the same pairs of vocalizers and reveals a small but significant correlation between the two. Discrepancies between classifier and behavioral performance could be due to (1) behavioral noise, (2) differences in the actual renditions for subject pairs that are used in the behavioral test vs. the classifier cross-validated dataset (both generated randomly) and (3) subjects emphasizing or using different acoustical features than the classifier. Note, however, that classifiers using spectrograms as feature spaces and nonlinear or noncontinuous decision boundaries yielded similar machine performance as those using PAF and LDA as reported in Fig. 3a (see Supplementary Fig. 1).

**Generalization across renditions from the same call type.** Since tests are composed of many trials, we tested whether subjects learn the discrimination by memorizing some aspect of each individual rendition or are able to generalize across renditions. Increases in behavioral performance over the course of

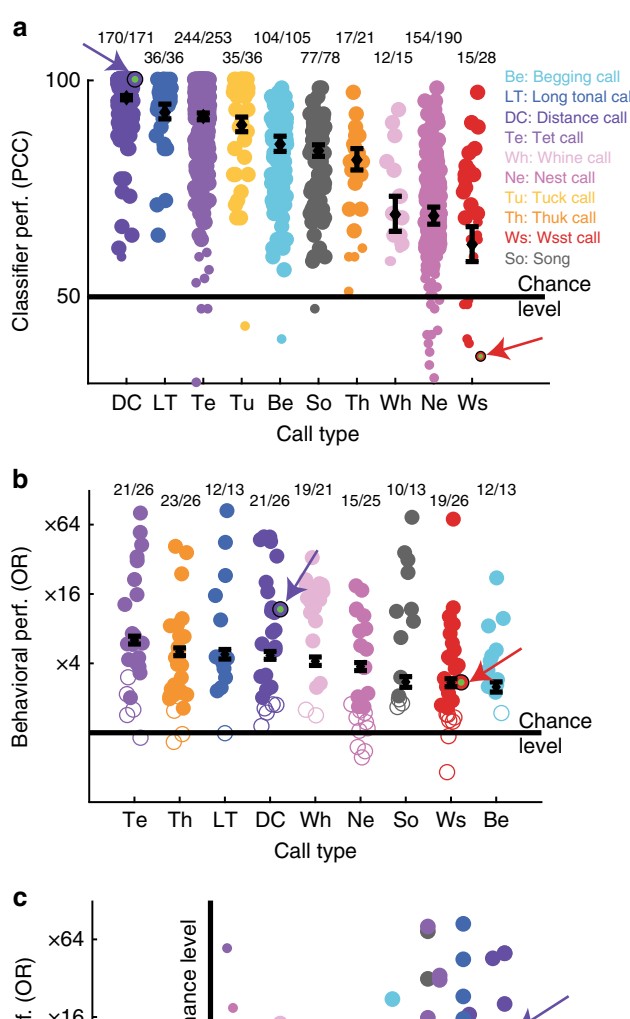

**Fig. 4** Vocalizer discrimination within each call type. **a** Each circle corresponds to the performance of the regularized LDA at discriminating a pair of vocalizers (large circles: acoustic discrimination of pairs of vocalizers significantly above chance tested by binomial test at $p < 0.05$ on PCC; effect of call type: likelihood ratio test (LRT) on Generalized Linear Mixed Effects models (GLME), $\chi^2_9 = 4406.5$, $p = 2.2 \times 10^{-16}$); **b** Each circle corresponds to the behavioral performance of bird subjects at discriminating pairs of vocalizers in a single-call-type test ($n = 13$). The behavioral performance is measured as the odds ratio, OR (filled circles: OR significantly above chance tested by exact test as described in Methods; effect of vocalizer: LRT on GLME, $\chi^2_1 = 2814.2$, $p = 0$; effect of call type: LRT on GLME, $\chi^2_8 = 700.7$, $p = 0$; see Supplementary Table 1 for all statistical results); **c** Correlation between the behavioral and the classifier performance for discriminating pairs of vocalizers (regression line indicated by the gray dotted line: $\rho = 0.2381$, $p = 0.0058$, two-tail Spearman coefficient, $n = 79$). The performances of the subject shown in Fig. 3 and of the classifier on the same stimuli are identified by arrows and small green dots in all three panels. Black diamonds and error bars in **a** and **b**: estimates of the GLME coefficients and their 95% confidence intervals. In all three panels, call type is color coded as described in **a**

single-call-type tests (Fig. 5a) demonstrate that birds learn to perform the discrimination task better later in the day for most call types (Likelihood ratio test or LRT on Generalized linear mixed effect model or GLME, $\chi^2_2 = 308.7$, $p = 0$; see Supplementary Table 1 for all statistical results). To test the hypothesis that improvement in discrimination was not due to birds learning and memorizing each individual rendition as being respectively rewarded or not rewarded, we examined the following predictions: (1) birds should discriminate above chance at the beginning of the daily test (first 30-min); (2) their behavioral performance should be significant on renditions they hear for the first time (which can be thought of as catch trials); and (3) for a given trial, their interruption behavior should not be better predicted by the number of times they have heard this particular rendition than the number of times they have heard this call type from this vocalizer. The behavioral data show that these three predictions are verified. First, birds discriminated vocalizers above chance in the first 30 min (Fig. 5a; LRT on GLME, $\chi^2_1 = 270.9$, $p = 0$). Second, birds scored higher than chance on stimuli that were heard only once or twice (Fig. 5b; LRT on GLME, $\chi^2_1 = 15.8$, $p = 7.0264 \times 10^{-5}$). Note that because these first and second renditions most likely appear during the first 30 min, showing a significant discrimination on these stimuli demonstrates generalization almost from the beginning of the test. Finally, the improvement in the correct behavioral response (interrupting NoRe and not interrupting Re) was fully predicted by the number of times birds have heard vocalizations from each vocalizer and there was no additional effect from the number of times they have heard each particular rendition (LRT on GLME, effect of VocRank $\chi^2_1 = 401.1$, $p = 0$; effect of RendRank with an offset based on VocRank; $\chi^2_1 = 3.2$, $p = 0.075258$). These results show that birds learn to generalize across renditions.

**Generalization across all call types.** After demonstrating the existence of individual signatures in every call type and their discriminability when tested independently, we investigated if birds could still identify vocalizers in a task that randomly mixed call types in the same test (all-call-type test, Figs. 6, 7). Despite the difficulty of the task given the number (Supplementary Table 2) and diversity of vocalizations, subjects tested with familiar vocalizers (Fig. 6a) achieved significant discrimination of same-sex vocalizers in 24 out of 26 tests (black dots in Fig. 6b) and generalized across renditions (Fig. 7a, b). Although these subjects had gained experience with the same vocalizers during the preceding single-call-type tests, this familiarity could not solely explain the results: similar behavioral performances were obtained with subjects naive to the pair of vocalizers (Figs. 6c, d; 7c, d and Supplementary Fig. 2). The behavioral performance depended on the call type and was significantly above chance for all types except Song (Figs. 6b, d; 7b, d). Song was also the least interrupted call type (Supplementary Fig. 3) and we hypothesize that the decrease in interruption reflects an intrinsic preference to listen to complete songs which interferes with the birds' performance.

Are birds using voice-cues shared between call types to perform this difficult task? To address this question, we investigated the generalization performance of classifiers to explore to what degree identity-bearing acoustic features are shared between call types. We found that the performance of the classifier dropped dramatically when it was trained and tested on different call types (Fig. 8). Similar effects are found when using other classifiers and/or the full spectrogram as an input feature space (Supplementary Fig. 1B). In summary, voice-cues are very weak in zebra finches. To achieve the significant discrimination of vocalizers in the very difficult task of the all-call-type tests, bird

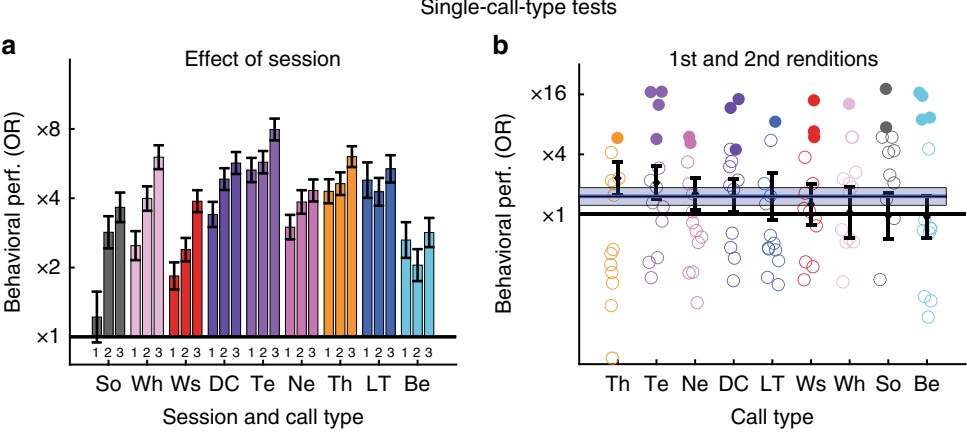

**Fig. 5** Behavioral generalization across renditions in single-call-type tests. Effect of gain of experience with the sounds over the course of a daily test on behavioral performance ($n = 13$ birds) where discrimination for vocalizer is assessed independently for each call type. **a** Behavioral tests were divided into three consecutive 30-min sessions labeled 1−3 (see Supplementary Methods). The bar plots illustrate the significant effect of session on bird performances. Call types are ordered along a decreasing effect of session from left to right. OR (odds ratio) increases with session rank for all adult call types but not for chick vocalizations (effect of Session:VocType: LRT on GLME, $\chi^2_2 = 308.7$, $p = 0$). Despite the behavioral improvement due to session training, the discrimination of vocalizers is above chance level even during the first 30-min session for all call types (effect of VocType on first session data: LRT on two GLME, $\chi^2_1 = 270.95$, $p = 0$). **b** The scatter plots show behavioral performances when the dataset is restricted to stimuli constructed with renditions heard once or twice. As in Fig. 4b, colored circles show performance of individual subjects and the black diamonds and error bars show the average performance and 95% confidence intervals obtained from the coefficients of GLME. Filled (vs. empty) circles indicate that the behavioral performance is significantly above chance. Birds significantly discriminate between the vocalizers achieving an average OR value across call types (blue line) with its 95% confidence interval (blue box) higher than chance level (effect of VocType on stimulus renditions heard once or twice: LRT on GLME, $\chi^2_1 = 15.8$, $p = 7.03 \times 10^{-5}$). There is also a significant interaction of OR with call type (black diamonds): Note that Te, Th, and DC are still among the best ID-discriminated categories even on this dataset restricted to first heard renditions (effect of the interaction VocType:CallType: LRT on GLME, $\chi^2_8 = 18.4$, $p = 0.018283$)

subjects might use some of the weak vocal-cues but, very likely, are also memorizing individual vocal signatures in particular for calls that are highly individualized (e.g. Tet, Songs, DC; Figs. 6 and 8). Taken together, these results demonstrate that birds can quickly (<100 trials) learn the set of signatures unique to each bird to identify vocalizers irrespective of the call type they produce.

## Discussion

Zebra finches can discriminate vocalizers based on any calls in the repertoire and do so by memorizing sets of individual signatures, one for each call type of the repertoire. Individual recognition across multiple call types could be useful, for example, to assess the reliability of the information provided by specific individuals[7] or to promote general collaborative or avoidant behavior with particular individuals.

Why have zebra finches evolved a system for individual recognition based on multiple signatures rather than passive-voice-cues that should be easier to produce, perceive and memorize? Voice-cues are the results, for a large part, of individual differences in the morphology of the vocal apparatus that affect all produced sounds. For example, differences in size, weight and age are reflected in the size of the vocal tract which, in turn, result in strongly individualized voice-cues (e.g. ref. [15].). In humans, the average spacing between formant frequencies across vowels is indicative of the sex and size of the individual. The variation in the vocal tract morphology between individuals is most likely the origin of that correlation[12,28]. For zebra finches, however, vocal tract filtering is not a reliable marker of identity. The individual variation of vocal tract length observed in adult zebra finches is between 28 and 33 mm and correspond to resonances in a very restricted frequency range of ~ 1 kHz around 5.5 kHz (or less with closed beak[50]). Not only is this range of passive resonance frequency small, but it also falls both within the range of resonant

frequencies (3−10 kHz) that can be actively modulated by changing the opening of the beak or the size of the oro-oesopharyngal cavity[51] and within the range used to encode distinct call types[31]. Indeed, zebra finches encode information in the spectral shape of their vocalizations with distinct peaks or formants ranging from 500 Hz to 8 kHz for each call type[31,52]. Contrary to humans, where speaker invariant vowel information is encoded mostly in the relative values between the first and second formant frequencies[28,53], zebra finches seem to code information with relatively precise frequency values of their resonant peaks[31]. We propose that because of the limited band-width of frequencies provided by small variations in size across animals relative to the variation produced across call types, what could be a reliable indicator of the bird size and individuality is masked by variation of the same acoustic parameter within a bird. This hypothesis provides an explanation on how the socio-ecological pressures of sounding different, while producing any call types in the repertoire and with the constraint of being a small animal, have pushed the individual signature in birds away from passive spectral shaping produced by the individualized morphology of their vocal organ. A similar hypothesis has been proposed for the evolution of identity bearing whistles in dolphins. They would use a signaling strategy based on signatures because passive-voice-cues become unreliable due to the pressure changing as the animals communicate at different depths[29].

If spectral shaping is not a reliable cue for identity, it is then expected that information about identity would rely more on temporal and fundamental frequency parameters than the information about call types. Performing a matched comparison of the information in spectral, temporal or fundamental features for discriminating vocalizers vs. call types, we show that the spectral feature space was the best subset of acoustical parameters for classifying call types (also in ref. [39]) and that temporal and fundamental features were further recruited for the

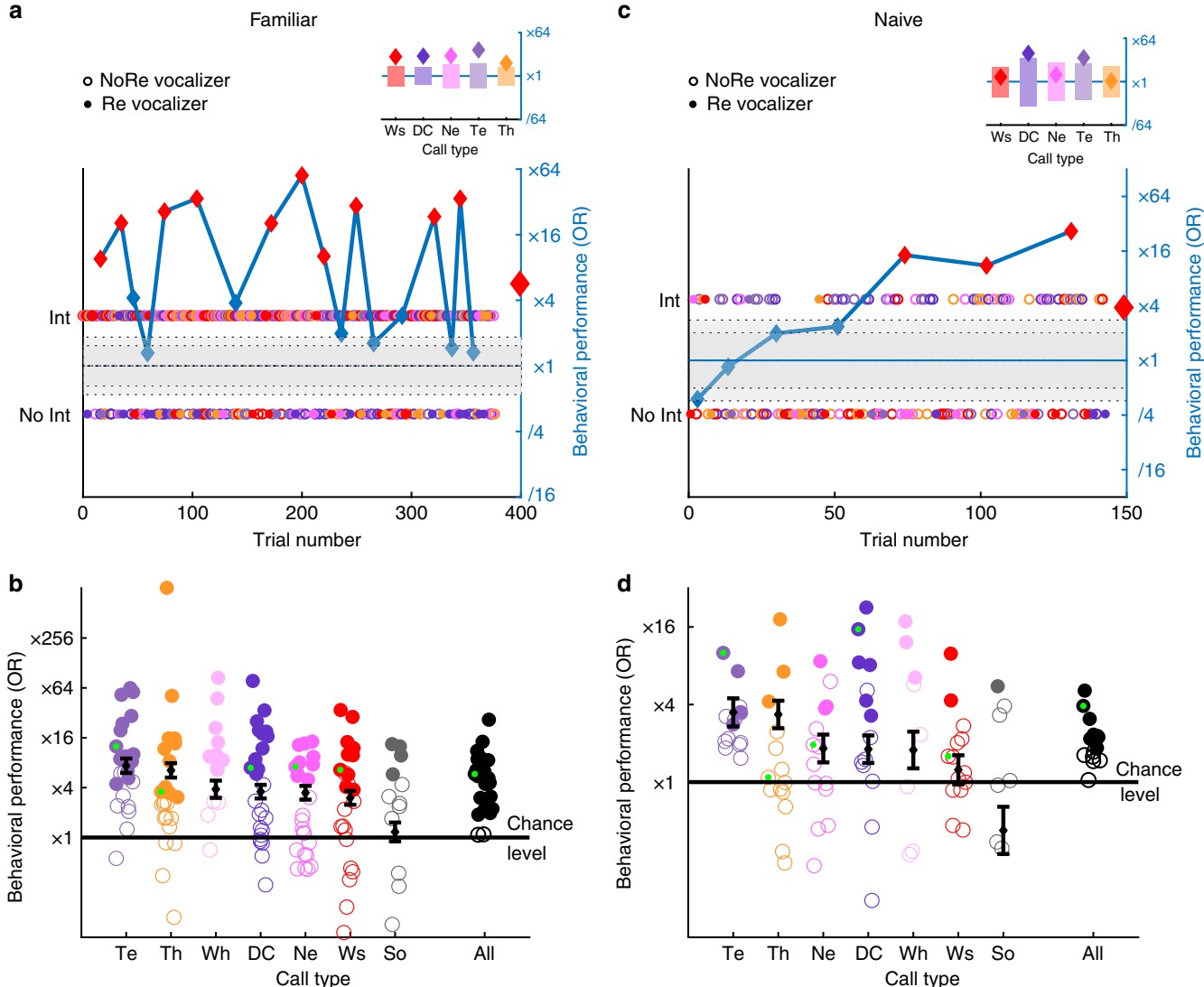

**Fig. 6** Behavioral generalization across call types. Familiar subjects (**a** and **b**) were tested in the all-call-type tests with the same vocalizers as in previous single-call-type tests. Naive subjects (**c** and **d**) performed the discrimination between vocalizers they had never heard before. **a−c** One bird subject performance at discriminating two female vocalizers across five call types (color coded as in Fig. 4a) and multiple renditions within each call type (all-call-type test; **a** $n = 313$ and **c** $n = 214$ renditions). Each dot corresponds to a stimulus. The behavioral performance as measured by the odds ratio (OR, see Fig. 1) is given by the blue line. Red diamonds indicate an OR value significantly different from 1. The gray areas depict the 5−95% and 1−99% quantiles for a random odds ratio for all trials. The insert shows the OR calculated for each call type across the entire test (diamonds) with the 5−95% confidence intervals for a random interruption behavior (bars). **b−d** Behavioral performance of all subjects (**b** $n = 13$; **d** $n = 7$) in the all-call-type test (effect of vocalizer, LRT on GLME: **b** $\chi^2_1 = 397.2$, $p = 0$, **d** $\chi^2_1 = 55$, $p = 1.2068 \times 10^{-13}$; effect of call type, LRT on GLME: **b** $\chi^2_6 = 247.7$, $p = 0$, **d** $\chi^2_6 = 172.4$, $p = 0$). Black diamonds and error bars correspond to the estimates of the GLME coefficients and their 95% confidence intervals. Filled circles indicate OR significantly above chance tested by exact test as described in Methods. Call type is color coded as described in Fig. 4a. The category All corresponds to the behavioral performance (OR) when all stimuli are considered, irrespective of the call type. OR performance for the subjects shown in **a** and **c** is identified by a small green dot in **b** and **d**

discrimination of vocalizers (Fig. 8c). This relative increase in the weight of fundamental and temporal features to encode identity information was particularly salient for contact and alarm calls (Supplementary Fig. 4). Analyzing the transfer of identity-bearing acoustic cues across call types for each subset of features shows that while temporal features mostly act as call-type-specific signature cues, spectral and fundamental features seem to be shared to some extent between types (voice-cues; Supplementary Fig. 4 and 5).

What are the mechanisms that generate this individual signature? Just as for spectral parameter variation, fundamental and temporal signatures could be due to the individual differences in

the morphology of the vocal organ, vocal tract or respiratory system. Yet, these acoustic features are the least correlated with morphological parameters[39]. A nonexclusive hypothesis is that these idiosyncratic differences are created by neural control. In this hypothesis, a set of individualized motor programs (one for each call type) would generate the temporal and fundamental variations carrying the vocalizer identity. While these individualized motor programs could be genetically inherited, vocal learning could further modify or overwrite these programs for some of the call types. For example, when learning their Distance call by imitating a model, male zebra finches overwrite their innate template while females do not[54]. It is highly probable that

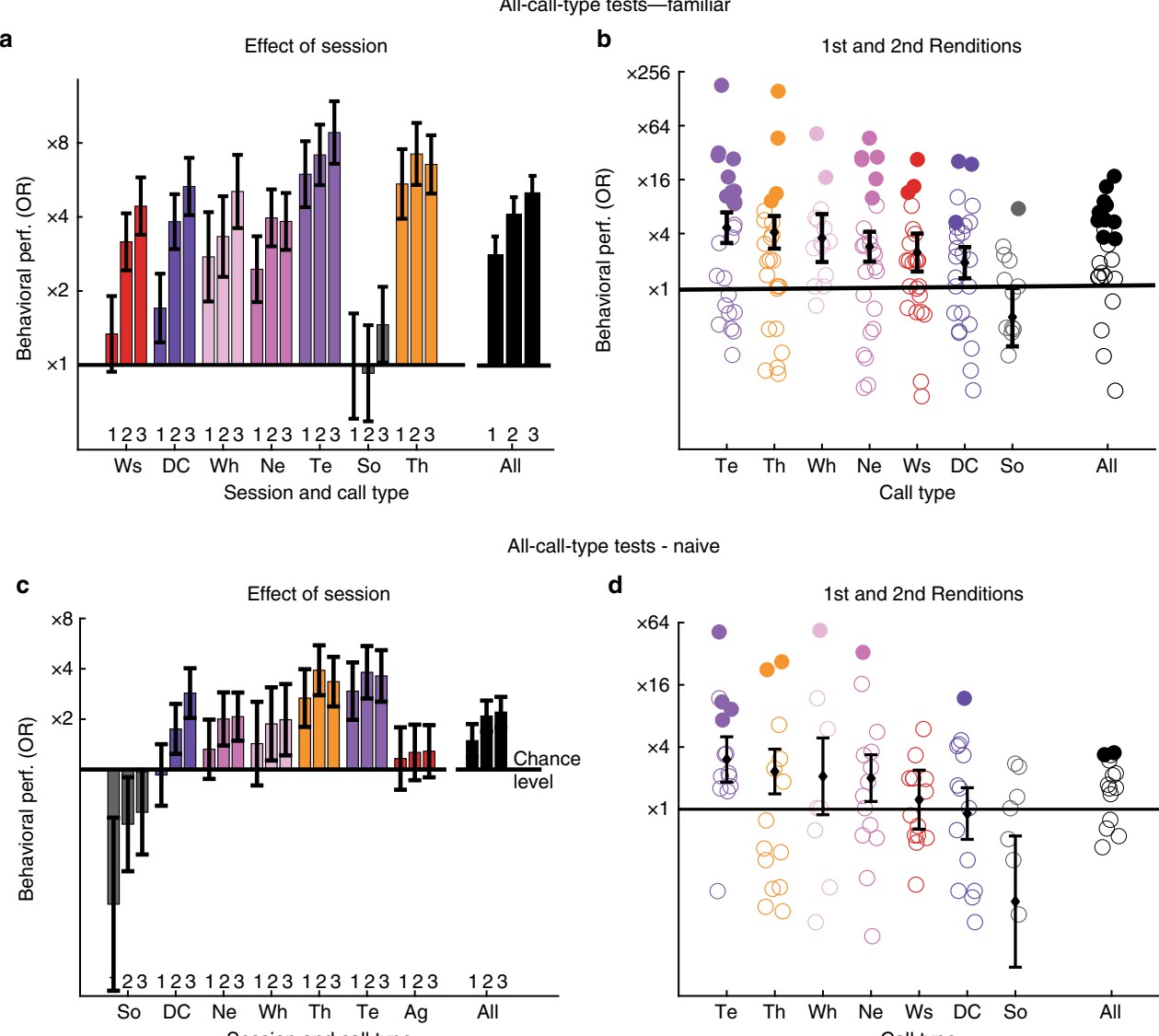

**Fig. 7** Behavioral generalization across renditions in all-call-type tests. Effect of gain of experience with the sounds over the course of a daily test on behavioral performance where discrimination for vocalizer is assessed for all call types simultaneously. **a**, **b** show the results when subjects ($n = 13$) are familiar to the vocalizers (same vocalizers as in single-call-type tests). **c**, **d** show the results when subjects ($n = 7$) are naive to the vocalizers. **a**, **c** Behavioral tests were divided into three consecutive 30-min sessions labeled 1–3 (see Supplementary Methods). Call types are ordered along a decreasing effect of session from left to right. For both Familiar and Naive subjects, the OR increases with session rank (effect of Session:VocType: LRT on GLME for Familiar, $\chi^2_2 = 59.5$, $p = 1.2035 \times 10^{-13}$; for Naive, $\chi^2_2 = 17.035$, $p = 1.9996 \times 10^{-4}$). **a** Despite the behavioral improvement due to session training, the discrimination of vocalizers is above chance level even during the first 30-min session for Familiar subjects (effect of VocType on first session data: LRT on GLME, $\chi^2_1 = 60.1$, $p = 8.9928e^{-15}$). The improvement in birds' performances depends on the call type (effect of Session:VocType:CallType: LRT on GLME, $\chi^2_{12} = 28.7$, $p = 0.0044348$). **b**, **d** The scatter plots show behavioral performances when the dataset is restricted to stimuli constructed with renditions heard once or twice. As in Fig. 4b, colored circles show performance of individual subjects and the black diamonds and error bars show the average performance and 95% confidence intervals obtained from the coefficients of GLME. Filled (vs. empty) circles indicate that the behavioral performance is significantly above chance. Vocalizations are labeled and color coded as in Fig. 4a. **b**, Familiar birds discriminate the vocalizers above chance level irrespective of the call type except on Songs (effect of VocType: LRT on GLME, $\chi^2_1 = 66.9$, $p = 3.3307 \times 10^{-16}$). Te and Th are the best ID-discriminated call types while Songs is not significant (effect of VocType:CallType: LRT on GLME, $\chi^2_6 = 52.3$, $p = 1.6413 \times 10^{-9}$; effect of CallType: LRT on GLME, $\chi^2_6 = 17.2$, $p = 0.0084594$). **d** Pairs of vocalizers were also discriminated on average above chance by subjects naive to the vocalizers (effect of VocType: LRT on GLME: $\chi^2_1 = 8.9$, $p = 0.002914$)

learning by imitation plays an important role in generating Distance calls that vary across birds while enforcing a very low within bird variability. Birds could also learn to avoid particular features that are already used as individual signatures by other familiar conspecifics. Further experiments comparing male and female vocal identity coding, and investigating the effect of learning could reveal the interplay between hardwired motor programs

and learned plastic motor programs. The crucial role that vocal learning in zebra finches could play in the production of highly individualized vocal signatures supports the recent hypothesis proposed by Pisanski et al.[28] to explain the origin of plasticity in early hominoid speech. Sounding different or similar to another individual could have motivated the evolution of the plastic control of vocalizations used by group-living animals[55] which, in

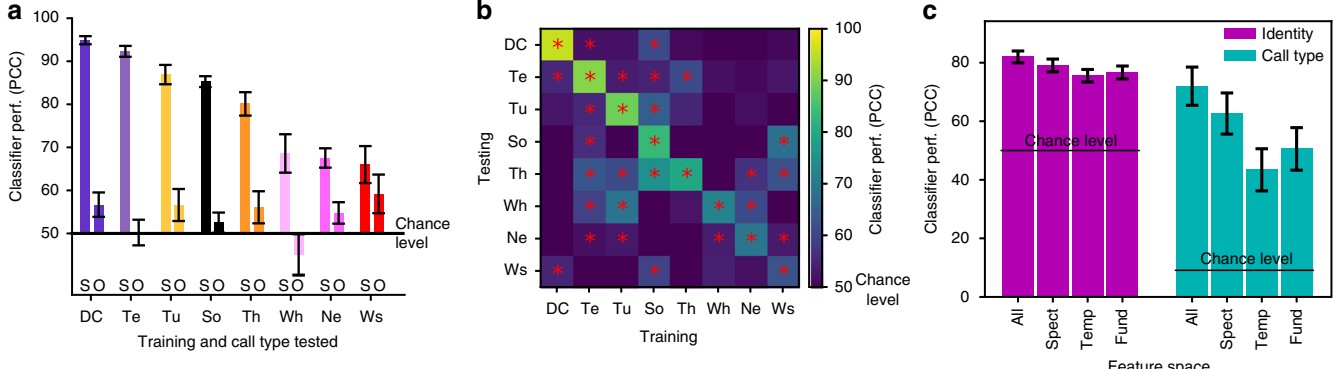

**Fig. 8** Classifier generalization across call types and acoustic features supporting classification. **a** Performance of the regularized LDA classifier at discriminating between pairs of vocalizers when trained either with a distinct set of renditions from the Same call type (S) or with the renditions of all Other four call types mixed together (O). If identity classification rules are shared by all call types, performance is expected to remain unchanged between S and O conditions. Performance drops drastically between Same and Other ($\chi^2_1 = 4710$, $p < 10^{-16}$) but remains above chance level for 5/8 call types. **b** Performance of the regularize LDA identity classifier on pair-wise sets of call types with the training call type indicated in columns and the tested category indicated in rows. Red stars indicate PCC significantly above 50% (binomial test, $p < 0.05$). The overall identity classification performance for these significant 24/56 cases was relatively poor (59 vs. 80% for the diagonal). **c** Classifier performance (PCC) for categorizing pairs of vocalizers averaged across all call types (Identity, purple) or classifying call types irrespective of the vocalizers (Call Type, green). The classifiers were trained on four different acoustic feature spaces: All, all 18 acoustic parameters (see Methods); Spect, 8 spectral parameters only; Temp, 5 temporal parameters only; Fund, 5 fundamental parameters only. For both Identity and Call Type classification there was a significant effect of the feature space but the effect size was much larger for Call Type than for Identity (Identity effect size: 28% increase in odds for the spectral features relative to fundamental or temporal; LRT on GLME: $\chi^2_2 = 853.76$, $p = 4.054496 \times 10^{-186}$; CallType effect size: 64% increase in odds for spectral features relative to fundamental and 118% increase in odds for spectral relative to temporal; LRT on GLME: $\chi^2_2 = 13.894$, $p = 0.0009614$; for call type analysis, acoustical features are averaged per bird across all renditions for a particular call type yielding smaller sample size). In **a** and **c**, error bars indicate 95% confidence intervals

humans, could have been further exploited to generate the linguistic content of speech sounds. The voice plasticity necessary to shape one's identity could be one of the crucial first steps in the evolution of language that could be shared across a few social and vocal vertebrates.

## Methods

**Subjects**. Fifty-eight domestic zebra finches (Taeniopygia guttata) were used for this study. Thirteen adults (7 females and 6 males; aged $11.5 \pm 2.3$ SEM month-old) were used as subjects for the behavioral experiments and 45 birds (adults: 13 females and 14 males; chicks: 7 females, 9 males and 2 of unknown sex, aged $20.3 \pm 0.9$ day-old) were used to constitute the vocalization databank used as stimuli (previously described in ref. [31]). Adult birds were issued from three different colonies housed either at UC Berkeley or UC San Francisco. All experiments were approved by the Animal Care and Use Committee at the University of California at Berkeley.

**Behavioral experiments**. To investigate the ability of adult zebra finches to discriminate vocalizers based on acoustic cues, we used a forced-choice operant conditioning paradigm previously developed in our laboratory and described elsewhere (see refs. [40,56], Fig. 1 and Supplementary Movie 1, custom Matlab code available upon request to the authors). Briefly, birds pecked a pad to induce the playback of sequences of vocalizations from two unfamiliar conspecific birds. Each acoustic stimulus consisted of a sequence of six and three band-pass filtered (0.25 −12 kHz) vocalizations of the same vocalizer and of the same call type, randomly assigned within a 6 s window. Vocalizations from one of the vocalizers were played infrequently (20% of the time) and rewarded by 10 s access to seeds that starts at the end of the ~6 s playback (Re Vocalizer) while the playback of vocalization sequences from the other vocalizer was frequent (80% of the trials) but not rewarded (NoRe Vocalizer). After initiating a trial (triggering the playback of a stimulus), the subject could either wait until the end of the playback or interrupt it by pecking again and, in this manner, initiate the following trial. The food access was only available after the playback of an Re Vocalizer call sequence and only if the subject waited until the end of the playback by refraining from interrupting: the interruption of the Re stimulus cancels the access to the reward. Because 80% of the trials would initiate the playback of NoRe Vocalizer call sequences while only 20% would initiate the playback of Re Vocalizer call sequences, subjects were expected to learn to specifically interrupt the NoRe Vocalizer and refrain from interrupting the Re Vocalizer in order to maximize their food access. This behavioral performance for the discrimination between the two vocalizers is quantified by the odds ratio (OR): the ratio between the odds of interrupting the NoRe vocalizer and the

odds of interrupting the Re vocalizer.

$$\mathrm{OR} = \log_2\left(\frac{O_{\mathrm{NoRe}}}{O_{\mathrm{Re}}}\right) = \log_2\left(\frac{p_{\mathrm{NoRe}}/(1-p_{\mathrm{NoRe}})}{p_{\mathrm{Re}}/(1-p_{\mathrm{Re}})}\right). \qquad (1)$$

Here $O_{\mathrm{NoRe}}$ is the odds of interrupting a NoRe stimulus, $O_{\mathrm{Re}}$, the odds of interrupting an Re stimulus, $p_{\mathrm{NoRe}}$, the probability of interrupting a NoRe stimulus and $p_{\mathrm{Re}}$, the probability of interrupting an Re stimulus. The probabilities of interruption were calculated by dividing the number of interrupted stimuli of a given type by the total number of triggered stimuli of that type. OR was calculated as a time running value or as an average value for all the trials of a given day.

To motivate the subjects, birds were fasted at least 15 h prior to the beginning of the experiment. Experiments started between 0900 hours and 1000 hours each day and terminated between 1400 hours and 1500 hours (~5 h duration). The birds were maintained in a fasting state (85−90% of their initial body weight) for the entire experimental series by giving them only 1.5 g of finch seeds at the end of each testing day on top of the rewards earned during the tests. The day light cycle was 0700−2100 hours. This conditioning paradigm does not use negative reinforcement for incorrect responses and its rules are quickly acquired by subjects; 3−4 h of training involving only a few hundreds of pecks are sufficient to learn the procedure[40,56].

For each subject, a random pair of males, a random pair of females, and a random pair of chicks were chosen from 24 vocalizers of our vocalization bank (7 females, 6 males, and 11 chicks). Each subject was then tested for its ability to discriminate these three pairs of vocalizers across all call types using different discrimination tasks. In the single-call-type tests, birds had to discriminate two male, two female or two chick vocalizers based on a single-call-type (7, 6, and 2 call types respectively, each tested on consecutive days; 15 tests total). In the all-call-type tests, birds had to discriminate two male or two female vocalizers based on all call types (up to 7 and 6 call types respectively tested at the same time; 2 tests total). To investigate the effect of the familiarity with vocalizations acquired during "single-call-type" tests on the behavioral performance of birds during "all-call-type" tests, seven female subjects run an additional set of "all-call-type" tests with vocalizers they had never heard before. Each test significance was tested by an exact test and the behavioral performance across subjects was evaluated with binomial Generalized Linear Mixed Effects models (GLME). See Supplementary Methods for further descriptions of the tasks and statistical analyses of behavioral performances.

**Acoustical analysis**. The acoustical analysis of vocalizations used the large database described in Elie and Theunissen[31] that contains vocalizations from 27 adults (13 females and 14 males) and 18 juvenile zebra finches, annotated for vocalizer ID and call type. To quantify the individual signature present in these vocalizations, we used three supervised classifiers: LDA, QDA, and RF. The classifiers were trained to sort

the vocalizations based on vocalizer ID using two distinct feature spaces to represent the sounds: PAF or an invertible spectrogram. The same classifiers based on the same PAF were also used to discriminate call types, repeating the analysis done in Elie and Theunissen[31], to compare the importance of specific PAF for discriminating vocalizers vs. call types. The PAF consisted of 18 features describing the spectral (8), temporal (5), and fundamental (5) characteristics of each sound (see Supplementary Methods and also ref.[31]). All acoustic features were obtained using custom Python code from the Theunissen lab (BioSound class in sound.py found in https://github.com/theunissenlab/soundsig; BioSound Tutorials with examples are found in https://github.com/theunissenlab/BioSoundTutorial).

All classifiers were trained and tested on separate data samples using up to tenfold cross-validation (min 5). Performance of the classifiers was quantified using percent correct classification on the cross-validated dataset. For the vocalizer classification shown in Fig. 3b for illustrative purposes, the classifier was trained to discriminate 11 vocalizers based on all the renditions from each of these birds for separate call types. For all other analyses on vocalizer discriminability, we trained and tested classifiers on all possible pair-wise comparisons. This pair-wise approach allowed us not only to directly compare the results to those obtained in the behavioral tests of vocalizer discrimination, but also to compare performance for classifying vocalizers across call types for which we had a different number of birds. Irrespective of the number of vocalizers, chance performance is based on pair-wise classification where 50% is chance level. Performance metrics for each pair of vocalizers and each classification test consisted in a binomial data (# of stimulus tested, # correctly classified) that was tested against chance using an exact binomial test. The significance of classifier performance across all vocalizers was tested by applying GLME on the number of cross-validated trials correctly classified vs. total number tested (see further details in Supplementary Methods).

Classification for vocalizers was estimated for each call type separately (with fitting and validation data coming from same call type) as well as for across call types. In the across-call-type classification, classifiers were tested on vocalizations drawn from one particular call type but either trained on vocalizations drawn from at least five out of the eight remaining adult call types ("Other" in Fig. 8a, Supplementary Fig. 1A and 5A) or from another call type (off-diagonal bins in Fig. 8b, Supplementary Fig. 1B and 5B). A minimum of five vocalizations from the same vocalizer per call type was required for that vocalizer to be included in these calculations.

## Data availability

The custom Python code used to calculate the acoustical features is available as the BioSound class in sound.py found in https://github.com/theunissenlab/soundsig. Tutorials with example data are found in https://github.com/theunissenlab/BioSoundTutorial. The complete library of vocalizations is available at Figshare: https://doi.org/10.6084/m9.figshare.11905533.v1. The behavioral data and behavioral analysis code are available from the corresponding authors upon reasonable request.

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

## Acknowledgements

We thank Nicolas Mathevon and Michael Yartsev for their comments on a previous version of this manuscript. We are also grateful to Michelle R. Carney, Timothy Day and Yuka Minton for their assistance with running behavioral experiments. This study was funded by the NSF IIS 131-1446 and the NIDCD R01-016783.

## Author contributions

Conceptualization, methodology, software, formal analysis, writing: J.E.E. and F.E.T.; Investigation: J.E.E.; Funding acquisition: F.E.T.

## Additional information

**Competing interests:** The authors declare no competing interests.

