## [Peer Review File · Nature Communications]

Reviewers' comments:

Reviewer #1 (Remarks to the Author):

Zebra finches are songbirds used in many studies of learning and vocal communication. Their vocal repertoire and behaviour has recently been documented in data-driven fashion. This paper provides evidence that zebra finches reared in lab conditions exhibit individual "signature" in their calls, in the sense that there is information embedded in the acoustic signal that can be used to discriminate individuals, and that other zebra finches can reliably detect this discriminative information. This confirms findings previously reported by the same lab and others. Further, the paper demonstrates that the effect is present across all call types in the zebra finch repertoire, with some call types more discriminable than others - and demonstrates that the distinctive features do not appear to come from uniform fixed "voice" character per individual, but from differences exhibited differently for each call type.

My expertise does not relate to the practical work in animal behaviour, but in the analysis of vocalisation data such as presented in this paper. Nevertheless, I believe the work is convincing and novel. The reproducibility is high due to the open publication of the analysis software and its clear documentation.

The acoustic features used for analysing vocalisations are good, and correctly chosen. The features are largely the same as the authors previously used, e.g. reference 28. In particular it is important to test both "PAFs" (predefined acoustic features) and spectrogram regressions (using RLDA, LDA, RF). Spectrogram regressions are good for finding discriminative differences if they exist, but are vulnerable to two possible weaknesses, namely overfitting to small datasets and of failing to detect correlations which are hard to detect due to time-warping of sounds. PAFs are much more robust to those two issues, but may fail to detect some information that is obscurely embedded in the signals. Thus the use of both feature sets is important for this work. The automatic classification results shown by the authors are supported by both analyses.

In Extended Fig 1 and elsewhere the authors show that the individual differences at the signal level are to a large extent call-specific and not global. This is also shown in the behavioural response data. This is the most significant finding of the work. It could be attributed to various causes, including genetic hard-coding, ontogeny, or learning. The authors argue in discussion that this is likely to come from learning, and an evolutionary need to transmit distinctiveness despite that "passive voice cues" may not provide enough audible variety.

The authors make clear (line 129-131) that the extent to which zebra finches discriminate among multiple individuals, and not merely their mate vs others, remains to be demonstrated. The present study demonstrates, for some calls in particular, this is possible in principle, for at least a modestly sized set of individuals.

Fig 3C illustrates that there is only an extremely weak relationship between the discrimination performance of the birds and of the automatic classifier.

There is an important issue of terminology, in the title as well as elsewhere (line 48). The presence of individual signature in a sound is not "name-like". To say something is "name-like" clearly implies the use of names as in human societies, where one person says "Hey Joe" and Joe reacts. The study here is about individual "signatures" not "names". This ambiguity has occurred before in popular press but we must be clear about the distinction. Authors must remove all use of the confused analogy with "name".

Minor issues, indexed by line or fig:

- * 40: "track" should be "tract"
- * 64 and elsewhere: "Despites" should be "Despite"
- * Fig 3: "Black diamonds" not really visible. "Black error bars?"
- * 103: "can be thought as" should be "can be thought of as"
- * Fig 4A and 4C: in the insert, the extent of the confidence intervals is very hard to see. Plot more clearly?
- * 201: "parameters" should be "parameter"
- * Suppl 199: is the representation really invertible? In typical spectrograms, the phase information is discarded which means the signal is not recoverable from the spectrogram.
- * Suppl 228: please give the software versions used, especially for scikit-learn which has some differences in implementation across versions.
- * The call-type names are not all spelt out in this paper (e.g. LT and Te on line 120). Please provide a table or other way for the reader to know the labels.

Reviewer #2 (Remarks to the Author):

I enjoyed reading this very interesting manuscript. The study tackles a very interesting question that has remained unanswered despite the plethora of articles on "vocal signatures" in animal vocal communication: is individual recognition supported by cues that are common to all call types, or instead by a set of cue unique to each call type?

Combining a set of elegant artificial classification and behavioural discrimination experiments, the authors provide a very convincing answer to this question in zebra finches. The methodology is exemplary and will serve as a benchmark for future investigations of this question in other vertebrate species. As such the article should have a considerable impact on the field.

The article is very clearly written and illustrations are sophisticated.

I only have a couple of minor comments:

- line 120: LT and Te are not defined in the text. More generally, acronyms for call types do not seem to follow clear rules ("call" included in acronym in DC but not in others) - though I appreciate that authors may use acronyms that have already been defined in the literature.
- line 166: "The variations in the vocal tract morphology between individuals is most likely the origin of that correlation" - Should it be "variation"? Or "are more likely". Throughout the manuscript, I wondered if "variations" should be singular.

Reviewer #3 (Remarks to the Author):

This study investigates whether zebra finches recognise other zebra finches individually based on their vocalizations only. This is investigated by using caller identity to create rewarded and unrewarded stimulus categories in operant discrimination tasks. Concurrent sound analyses try to identify candidate individual markers in individual sound repertoires.

1) This study is an interesting investigation of the extent of individual recognition in a bird species that is an important model in communication studies but the authors overstate the significance of their findings: There are several published studies on vocalisation mediated individual recognition for different call types in this species (e.g. Fernandez et al. 2016, Kniel et al. 2016, Perez et al. 2016, Butler et al. 2017, Ma et al. 2017 to name just a few that appeared in the past two years) which means that the knowledge gain arising from the manuscript is incremental rather than novel. The

argument that there has never been a full repertoire tested is a bit of a strawman (btw are the authors now testing a full repertoire of each tested individual? And do all human vocalizations carry an individual signature? I thought whispering, screams of highest anguish and agony cannot be assigned to individuals either?).

2) The introduction discusses the interesting issue of active versus passive voice cues but implies that the study is addressing this issue but do the behavioural results allow to conclude which mechanisms underly the individual recognition?

3) The manuscript is 54 pages long and the way it is currently organised it is very difficult to follow. Parts of the methods are in the figure legend, some in the method section and the rest in the supplement but they are crucial to understand the results. I appreciate the problem arising from the journal format asking the results to be presented before the methods, but even with these constraints there might be ways to have the crucial aspects of the methods all in one place (perhaps making them more compact by using a table for the timeline/associated tests/stimuli? Also, if you could formulate clear predictions regarding each test (if birds can do A we expect that they to perform xyz with test sounds xyz..).

4) Figures: please check that all abbreviations used in a figure are explained and that the figure legends explain what data are shown – in several figures, I could not figure what the individual dots meant.

5) Please provide more detail on the stimulus presentations: I read the manuscript now 2 times and I am still not sure how many test sounds and probe sounds were there per individual (and in total?) and how the 6s stimuli were constructed (how much sound vs. silence, how many call repetitions/s) Did all individuals get different or the same test sounds for a particular call category? How long took the training/testing (on average, range) last and were birds housed in the testing setup or each returned to their home cage? Was their food deprivation before each test or only before the initial training? How many hours per day did the bird have access to food in the training? In which order did individuals experienced the tests (I get the impression they all got the same order?). Nomenclature for zebra finch calls is not the same across studies – it would be very helpful to have a table with the call types belonging to the abbreviations used in the figure, a brief definition and a representative spectrogram.

Overall, the Nature Communications format (methods after the results) seems poorly suited for this particular manuscript because to understand the results one needs to know the methods, to understand the methods one needs to read the supplement – and after 54 pages manuscript text it is still not clear to me exactly what was tested and how. Tables with stimulus information, and visualisation of the procedure (for example flow charts would help), as is the methods provide not enough information for replication and the manuscript will be difficult to read by a wider audience.

Butler NE et al. 2017: Lack of alarm calls in a gregarious bird: models and videos of predators prompt alarm responses but no alarm calls by zebra finches. *Behav Ecol Sociobiol* 71: Unsp 113.

10.1007/s00265-017-2343-z

Fernandez MSA et al. 2016: A New Semi-automated Method for Assessing Avian Acoustic Networks Reveals that Juvenile and Adult Zebra Finches Have Separate Calling Networks. *Frontiers in Psychology* 7: 1816. 10.3389/fpsyg.2016.01816

Kniel N et al. 2016: Sex-Specific Audience Effect in the Context of Mate Choice in Zebra Finches. *Plos One* 11: e0147130. 10.1371/journal.pone.0147130

Ma SW et al. 2017: Power-law scaling of calling dynamics in zebra finches. *Scientific Reports* 7: 8397. 10.1038/s41598-017-08389-w

Perez EC et al. 2016: Corticosterone triggers high-pitched nestlings' begging calls and affects parental behavior in the wild zebra finch. *Behav Ecol* 27: 1665-1675. [10.1093/beheco/arw069](https://doi.org/10.1093/beheco/arw069)

Dear Reviewers,

We thank you for your work and comments. We have attempted to address all of your criticism as described in detail below in response to each of your queries. Our responses are in black font and italicized.

Thanks again for your help.

Best,

Julie Elie and Frederic Theunissen.

Reviewer #1 (Remarks to the Author):

Zebra finches are songbirds used in many studies of learning and vocal communication. Their vocal repertoire and behaviour has recently been documented in data-driven fashion. This paper provides evidence that zebra finches reared in lab conditions exhibit individual "signature" in their calls, in the sense that there is information embedded in the acoustic signal that can be used to discriminate individuals, and that other zebra finches can reliably detect this discriminative information. This confirms findings previously reported by the same lab and others. Further, the paper demonstrates that the effect is present across all call types in the zebra finch repertoire, with some call types more discriminable than others - and demonstrates that the distinctive features do not appear to come from uniform fixed "voice" character per individual, but from differences exhibited differently for each call type.

My expertise does not relate to the practical work in animal behaviour, but in the analysis of vocalisation data such as presented in this paper. Nevertheless, I believe the work is convincing and novel. The reproducibility is high due to the open publication of the analysis software and its clear documentation.

The acoustic features used for analysing vocalisations are good, and correctly chosen. The features are largely the same as the authors previously used, e.g. reference 28. In particular it is important to test both "PAFs" (predefined acoustic features) and spectrogram regressions (using RLDA, LDA, RF). Spectrogram regressions are good for finding discriminative differences if they exist, but are vulnerable to two possible weaknesses, namely overfitting to small datasets and of failing to detect correlations which are hard to detect due to time-warping of sounds. PAFs are much more robust to those two issues, but may fail to detect some information that is obscurely embedded in the signals. Thus the use of both feature sets is important for this work. The automatic classification results shown by the authors are supported by both analyses.

In Extended Fig 1 and elsewhere the authors show that the individual differences at the signal level are to a large extent call-specific and not global. This is also shown in the behavioural response data. This is the most significant finding of the work. It could be attributed to various causes, including genetic hard-coding, ontogeny, or learning. The authors argue in discussion that this is likely to come from learning, and an evolutionary need to transmit distinctiveness despite that "passive voice cues" may not provide enough audible variety.

The authors make clear (line 129-131) that the extent to which zebra finches discriminate among multiple individuals, and not merely their mate vs others, remains to be demonstrated. The present study demonstrates, for some calls in particular, this is possible in principle, for at least a modestly sized set of individuals.

Fig 3C illustrates that there is only an extremely weak relationship between the discrimination performance of the birds and of the automatic classifier.

We agree that this correlation is small (but maybe not extremely weak ;-): $Rho = 0.2$ is small to medium effect size according to social science conventions). To be succinct, we had omitted a section discussing this relationship in the initial manuscript. We now briefly discuss this result in the main text and the multiple potential reasons for the small effect. Primarily, we believe that behavioral data is relatively noisy. The new paragraph starts line 122 with: "We also investigated if the difference in behavioral performance..."

There is an important issue of terminology, in the title as well as elsewhere (line 48). The presence of individual signature in a sound is not "name-like". To say something is "name-like" clearly implies the

use of names as in human societies, where one person says "Hey Joe" and Joe reacts. The study here is about individual "signatures" not "names". This ambiguity has occurred before in popular press but we must be clear about the distinction. Authors must remove all use of the confused analogy with "name".

We agree with the reviewer and eliminated the two references to "name" that were in the original manuscript. We replaced the example for labels on line 48 by the human laughter that can also have a label-like individual signature.

Minor issues, indexed by line or fig:

* 40: "track" should be "tract"

Corrected in the revised version

* 64 and elsewhere: "Despites" should be "Despite"

Corrected in the revised version line 68, 134 and in the supplemental methods

* Fig 3: "Black diamonds" not really visible. "Black error bars?"

Corrected for "black diamonds and error bars" in the legend of figure 4 of the revised version (ex-figure 3).

* 103: "can be thought as" should be "can be thought of as"

Corrected line 93 in the revised version

* Fig 4A and 4C: in the insert, the extent of the confidence intervals is very hard to see. Plot more clearly?

We changed the insert such that only 5-95% confidence intervals appear. We believe that the confidence intervals are more visible in what is now figure 5 in the revised version.

* 201: "parameters" should be "parameter"

Corrected in the revised version line 200.

* Suppl 199: is the representation really invertible? In typical spectrograms, the phase information is discarded which means the signal is not recoverable from the spectrogram.

It is true that many spectrograms are not invertible as the phase of the short-time Fourier Transform is discarded. However, the phase of the actual signal (and not the windowed signal) can be recovered given certain constraints on the shape of the window and the overlap between consecutive windows.

*A very simple proof can be found in the excellent text by Leon Cohen: "Time frequency analysis: Theory and applications" (see Chapter 7, 7.9 p. 108). The parameters we are using in the present manuscript enable almost perfect recovery of the phase and thus the original waveform. We have performed such tests in Singh and Theunissen (2003) for zebra finch song and in Elliot and Theunissen (2009) for speech. We changed the wording in the manuscript to "we also used a **practically** invertible spectrographic representation" and added the citation to Singh and Theunissen.*

* Suppl 228: please give the software versions used, especially for scikit-learn which has some differences in implementation across versions.

We added this precision the revised version of the supplemental methods.

* The call-type names are not all spelt out in this paper (e.g. LT and Te on line 120). Please provide a table or other way for the reader to know the labels.

To guide the reader to the fully spelt-out versions of the calls shown in Fig4 (ex figure 3), we added "see Fig 4 for all call labels" on the first occurrence of the labels line 110-111.

Reviewer #2 (Remarks to the Author):

I enjoyed reading this very interesting manuscript. The study tackles a very interesting question that has remained unanswered despite the plethora of articles on "vocal signatures" in animal vocal communication: is individual recognition supported by cues that are common to all call types, or instead by a set of cue unique to each call type?

Combining a set of elegant artificial classification and behavioural discrimination experiments, the authors provide a very convincing answer to this question in zebra finches. The methodology is exemplary and will serve as a benchmark for future investigations of this question in other vertebrate species. As such the article should have a considerable impact on the field.

The article is very clearly written and illustrations are sophisticated.

I only have a couple of minor comments:

- line 120: LT and Te are not defined in the text. More generally, acronyms for call types do not seem

to follow clear rules ("call" included in acronym in DC but not in others) - though I appreciate that authors may use acronyms that have already been defined in the literature.

Indeed, we decided to keep already existing acronyms in the field and decided to stay consistent with our description of the repertoire as described in the Animal Cognition manuscript (Elie and Theunissen, 2016). We agree that the spelt-out versions of the call types were not easy to find. To guide the reader to the fully spelt-out versions of the calls shown in Fig4 (ex-fig3), we added "see Fig 4 for all call labels" on the first occurrence of the labels line 110-111.

- line 166: "The variations in the vocal tract morphology between individuals is most likely the origin of that correlation" - Should it be "variation"? Or "are more likely". Throughout the manuscript, I wondered if "variations" should be singular.

We agree with the reviewer that variation should have been singular in many places and changed it when appropriate throughout the manuscript, or replaced by "differences".

Reviewer #3 (Remarks to the Author):

This study investigates whether zebra finches recognise other zebra finches individually based on their vocalizations only. This is investigated by using caller identity to create rewarded and unrewarded stimulus categories in operant discrimination tasks. Concurrent sound analyses try to identify candidate individual markers in individual sound repertoires.

1) This study is an interesting investigation of the extent of individual recognition in a bird species that is an important model in communication studies but the authors overstate the significance of their findings: There are several published studies on vocalisation mediated individual recognition for different call types in this species (e.g. Fernandez et al. 2016, Kniel et al. 2016, Perez et al. 2016, Butler et al. 2017, Ma et al. 2017 to name just a few that appeared in the past two years) which means that the knowledge gain arising from the manuscript is incremental rather than novel.

We agree with the reviewer that there have been papers investigating individual signatures in the vocalizations of the zebra finch – although not all the references listed here by the reviewer address specifically this issue. As I am sure that the reviewer and editor can assess, we were very upfront in our text with the fact that, in zebra finches, individual recognition for particular calls had been demonstrated and included 10 very relevant citations:

"Because the coding of identity has been shown independently for some of these call types (Begging calls, (30, 31); Distance calls (1, 32-34); Song (35-37) and soft calls (38)), it is a good model to investigate mechanisms of identity coding through a whole repertoire."

The previous version was a direct transfer from another journal which was limited in terms of number of citations. In the revised version, we put back 8 more citations that are also related to the investigation of identity coding in zebra finches, and the same sentence can now read:

"Because the coding of identity has been shown independently for some of these call types (Begging calls, (31, 32); Distance calls (1, 30, 33-39); Song (40-43) and soft calls (44, 45) but see (46) for Long Tonal calls)), it is a good model to investigate mechanisms of identity coding through a whole repertoire."

As state, we believe that these prior finding constitute a strong scientific premise for the investigation of voice in zebra finches pursued here. Voice investigation requires the assessment of the recognition of the individuality across an entire repertoire; meaning that the signature of the same individuals must be assessed across all call types. This had not been investigated in zebra finches species nor in any other species as far as we know. In most ethological studies, investigation of individual recognition in animals has focused on demonstrating individual discrimination using a single call type and finding the acoustic cues that carry the identity of the vocalizer in that category. We were in a unique position of having audio recordings of entire repertoires from many individuals and thus of being able to investigate how individual information could be conveyed throughout a repertoire. In particular, we were able to test whether individual information was carried by voice qualities that generalized across all call types. This is an important question that has ramifications both for perceptual and for motor

mechanisms involved in vocal communication. For these reasons, we respectfully disagree with the assessment that our contribution is incremental.

In the paper, we succinctly (due to page limitations) describe the most relevant prior work in Zebra Finches but this does not imply that we are not very well aware of the body of work performed in this species. The repertoire of the zebra finch is organized along 11 call types according to the meaning or social context in which they are emitted (1). Four of these categories have received some attention regarding kin recognition. The song of the male, by far the call type that has been most studied, is highly individualized and conveys the identity of males that is decoded by both males and females conspecifics (2-5). The second most studied vocalization of the zebra finch is a long-distance contact call, the Distance call. Emitted by both adult males and females, this call type is also clearly individualized, conveying information about the sex and the identity of the vocalizer (6-11). Mates rely on this contact call to recognize each other when out of sight (6, 7, 12) and young birds use it to recognize their parents (13, 14). Young birds also emit a contact call that will progressively mature to a Distance call, the Long Tonal call (1, 12). Juveniles use this vocalization to establish contact when out of sight of their parents or siblings. While this vocalization bears enough individualized acoustic features to enable a classification of vocalizers above chance by algorithms, play-back experiments in parents failed to demonstrate a behavioral discrimination of young based on that call type (15). Finally, because young have been observed to continue to beg for food after fledging and join their nest-mates to collectively beg to their parents, the Begging call was also suspected to carry information about identity; Levrero et al demonstrated that nestlings emit acoustically individualized begging calls that can be recognized by parents (16). Ligout et al further showed that after fledging, juveniles continue to emit acoustically individualized begging calls that they can use to recognize their nest-mates (17). In addition to these 4 call types, the repertoire of the zebra finch includes other categories (1, 12) that have received little if any attention regarding individual recognition. Some of these vocalizations are used for communication at short distance between familiar or mated birds (the short distance contact call Tet, the Nest calls, the Whine; (18-20)), others are used in agonistic interactions (the aggressive call Wsst and the distress call; (1, 12)) and 2 are used to sound the alarm (the Tuck call and Thuk call; (1, 12)). Play-back experiments using the soft vocalizations (e.g. Tet, Nest calls) pointed to a recognition of mates based on these call types but also called for further explorations (18, 21). While for half of the call types, previous research have demonstrated a coding of individual signature, the individuality across the entire repertoire has never been investigated.

The argument that there has never been a full repertoire tested is a bit of a strawman (btw are the authors now testing a full repertoire of each tested individual?)

As explained above, we don't believe that this argument is a strawman. We are not sure how one would be able to test for a voice signature otherwise. And yes, we are testing almost the full repertoire of each individual with small exceptions that are principally due to biological limitations. The zebra finch repertoire contains 11 call types as described in Elie and Theunissen 2016. Two of these call types constitute the repertoire of chicks (LT and Be). Among the 9 remaining call types, one is only uttered by males (So) and the 8 others are uttered by both males and females. The distress call (Di) and the alarm call (Tuck, Tu) could not be tested in the behavioral paradigm as birds would freeze and avoid pecking on the key pad to hear these vocalizations. As explained line 260 onwards in the method section bird subject were tested on all possible call types for a given vocalizer of a given sex and age: male vocalizers had 9-2=7 call types, female vocalizers had 8-2=6 call types and young vocalizers had 2 call types. Line 262 reads:

"Each subject was then tested for its ability to discriminate these 3 pairs of vocalizers across all call types using different discrimination tasks. In the single-call-type tests, birds had to discriminate 2 male, 2 female or 2 chick vocalizers based on a single call type (7, 6 and 2 call types respectively, each tested on consecutive days; 13 tests total). In the all-call-type tests, birds had to discriminate 2 male or 2 female vocalizers based on all call types (up to 7 and 6 call types respectively tested at the same time; 2 tests total)."

And do all human vocalizations carry an individual signature? I thought whispering, screams of highest anguish and agony cannot be assigned to individuals either?).

To the best of our knowledge, this question has not been well investigated for non-speech sounds. There is clearly individual signature in cries and in laughter (See for example: Gustafsson et al Fathers

are just as good as mothers at recognizing the cries of their baby, *Nature Communication* 4 1698, 2013 and citation 28 in the paper: Bachorowski JA, Smoski MJ, & Owren MJ (2001) *The acoustic features of human laughter. Journal of the Acoustical Society of America* 110(3):1581-1597.). This is an active area of research in the field of human emotion and communication and we believe that our analyses and results will be of great interest to that community as well.

2) The introduction discusses the interesting issue of active versus passive voice cues but implies that the study is addressing this issue but do the behavioural results allow to conclude which mechanisms underly the individual recognition?

We agree with the reviewer that the behavioral results do not allow to conclude on the mechanisms used by zebra finches to discriminate individuals. Rather, the behavioral results demonstrate that they are able to achieve this difficult task irrespective of the call type. However, the acoustic analysis that we conducted demonstrates that the individual information is encoded using different acoustic parameters depending on the call type and that these parameters cannot correspond to passive filtering or passive voice cues. As a consequence, and with little doubt, birds discriminate individuals based on acoustic parameters that are under active control.

3) The manuscript is 54 pages long and the way it is currently organised it is very difficult to follow. Parts of the methods are in the figure legend, some in the method section and the rest in the supplement but they are crucial to understand the results. I appreciate the problem arising from the journal format asking the results to be presented before the methods, but even with these constraints there might be ways to have the crucial aspects of the methods all in one place (perhaps making them more compact by using a table for the timeline/associated tests/stimuli? Also, if you could formulate clear predictions regarding each test (if birds can do A we expect that they to perform xyz with test sounds xyz..).

As admitted by the reviewer, it is difficult to completely reorganize how methods are presented in the manuscript. The reviewer would like a short and long version of the methods sections so that the results can be understood without going back and forth to the Methods and Supplemental Material. This is a reasonable request. We have attempted to further explain the experimental procedure right at the start of the manuscript. For this purpose, we are relying on an additional figure, Fig2, and additional text in that legend. The new panel clearly shows the time course of the experiments for each bird and, thus, all the behavioral tests that are performed in this study. The legend gives additional details on the methods. The reader is exposed to that figure at the very beginning of the manuscript and can then refer to the methods sections and additional material for particular details (and we added many more as requested by this reviewer and as described below). The predictions of the behavioral tests are listed in the main text (and not in a table as suggested) but we believe that they are going to be better understood now that the reader has the global view of the behavioral tests.

4) Figures: please check that all abbreviations used in a figure are explained and that the figure legends explain what data are shown – in several figures, I could not figure what the individual dots meant.

We guess that the reviewer is referring to the call category abbreviations that was given in figure 4 (ex-figure 3) but not well referenced in the text. This is now resolved as mentioned above. Concerning dots in figures, we added details in figure 1, 3 (ex-2) and 5 (ex-4) to precise what the dots represent.

5) Please provide more detail on the stimulus presentations: I read the manuscript now 2 times and I am still not sure how many test sounds and probe sounds were there per individual (and in total?)

*The supplementary methods table 1 displays the number of call iterations of each vocalizer used on average over tests and subjects for each type of discrimination. In the supplementary methods, line 52 we also indicate that 5-104 (37.7 ± 1.4) different renditions per vocalizer and per call type were used per test. Line 52-53 of the supplemental material, we added the sentence: **“A total of 3283 vocalizations were used for these experiments”**.*

and how the 6s stimuli were constructed (how much sound vs. silence, how many call repetitions/s)

The construction of stimuli is described in details in the supplementary methods with added details shown in bold below:

“Each acoustic stimulus consisted of a sequence of six or three band-pass filtered (0.25-12 kHz) vocalizations of the same vocalizer and of the same call type, randomly assigned within a 6s window. More precisely, for the longer Begging sequences and Songs, each stimulus consisted of sequences of 3 different renditions, while for the other call types (Distance call, Nest call, Tet call, Tuck call, Whine call, Wsst call and Long Tonal call) each stimulus consisted of 6 different renditions. Before each session, the computer was randomly constructing a minimum of 80 Re stimuli and 320 NoRe stimuli using a vocalization bank of 5-104 (37.7 ± 1.4) different renditions per vocalizer and per call type (see supplementary methods Table 1). **A total of 3283 vocalizations were used for these experiments. Each of the 400 stimuli (i.e. sequences of six or three different renditions) was only played once during the session.**”

The balance between sound and silence depends on the vocalization type, indeed, some vocalizations are shorter (e.g. thuk call ~70ms) than others (e.g. begging call sequence ~1.8s). To give the reader an idea of the variation of the balance between silence and sound for a given type of test, we added a column to Supplementary methods table 1 that indicates the average, over stimuli, of the total duration of silence in a stimulus for each type of test.

Did all individuals get different or the same test sounds for a particular call category?

As indicated in the methods: “For each subject, a random pair of males, a random pair of females and a random pair of chicks were chosen from 24 vocalizers of our vocalization bank (7 females, 6 males and 11 chicks). Subjects were then tested for their ability to discriminate these vocalizers across all call types using the following 6 different types of discrimination tasks (supplementary methods Table 1).”

Given the number of subjects (13) and the number of individuals from which to choose in each category of vocalizers (7 females, 6 males and 11 chicks), some subjects by chance did end up working with the same pair of vocalizers.

How long took the training/testing (on average, range) last and were birds housed in the testing setup or each returned to their home cage?

The supplementary methods indicated that the shaping phase lasted less than 5 days. We now indicate the range: “Birds were shaped to use the operant chamber over a short period of time (**2-5 days**) using two songs from different male zebra finches as Re and NoRe stimuli.”

The testing phase consisted in a total of 7 Male-single-call-type + 1 Male-all-call-type + 6 Female-single-call-type + 1 Female-all-call-type + 2 Chick-single-call-type + 1 Random tests = 18 days of testing corresponding to 18 different tests as described in the supplementary methods, as shown in supplementary methods table 1 (number of lines in the table, not including the All-F2 and All-M2 tests that were done only for the female subjects) and now as depicted in figure 2.. Each subject run these tests in 3 series of consecutive days of testing: tests on male vocalizations (7 Male-single-call-type + 1 Male all-call-type), tests on female vocalizations (6Female-single-call-type + 1 Female-all-call-type) and tests on chicks vocalizations and the random test (2Chick-single-call-type + 1Random. Figure 2 gives a clear description of the series of tests and the supplemental methods describes line 111-112: “All tests were performed in series of maximum 10 consecutive days and always started **after** a shaping day (Day 0).”

To state more clearly that birds always returned to their home cage in the colony room we added details indicated in bold in the following sentence that can be read in the supplementary methods line 11-12: “For the duration of the shaping and testing days and while not **in the testing chamber**, the subjects were housed individually or in pairs **in the colony room** and fasted: their food intake was fixed to 1.5g of mixed seeds for finches per individual and was given at the end of each day upon returning to the colony room.”

Was their food deprivation before each test or only before the initial training?

Birds were deprived for both as indicated in this sentence of the supplementary methods line 11-12: "For the duration of the shaping and testing days and while not **in the testing chamber**, the subjects were housed individually or in pairs **in the colony room** and fasted: their food intake was fixed to 1.5g of mixed seeds for finches per individual and was given at the end of each day upon returning to the colony room." and in this sentence from the methods section line 254-257: "To motivate the subjects, birds were fasted at least 15 hours prior to the beginning of the experiment and maintained in a fasting state (85-90% of their initial body weight) for the whole experiment by giving only 1.5g of finch seeds at the end of each testing day on top of the rewards earned through the tests."

How many hours per day did the bird have access to food in the training?

When in the chamber (4.5 hours), birds had access to food whenever they brought the feeder up by being successful at the task during the 3*30min sessions. The number of times they succeed at the task varied from one bird and one day to the other. The food access at each success was 10 seconds as indicated in the method section of the article line 235-239: "Vocalizations from one of the vocalizers were played infrequently (20% of the time) and rewarded by ten second access to seeds that starts at the end of the ~6s playback (Re Vocalizer)".

In which order did individuals experienced the tests (I get the impression they all got the same order?).

The order in which the individuals experienced the tests is indicated in the supplementary methods: "Note that vocalizer single-call-type discrimination tests were always performed before vocalizer all-call-type discrimination tests. For the vocalizer single-call-type discrimination tests, the order in which call types were tested was randomly assigned to each subject". We also think that the Figure 2 with its legend is bringing more details on that aspect.

Nomenclature for zebra finch calls is not the same across studies – it would be very helpful to have a table with the call types belonging to the abbreviations used in the figure, a brief definition and a representative spectrogram.

The abbreviations used for the vocalization types are indicated in figure 4 (ex-figure3). Two of the categories are shown as examples in Figure 3 (ex-figure2). The full bank is made available so anyone will be able to investigate the spectrograms of any call. Finally, the full descriptions and spectrograms of all the categories investigated here can be found in the article that first described the database of sounds used in the present study: Julie E Elie and Frédéric E Theunissen. 2015. The vocal repertoire of the domesticated zebra finch: a data-driven approach to decipher the information-bearing acoustic features of communication signals. *Animal Cognition*. It is indicated in the manuscript line 227-228: "**45 birds (20 females, 23 males and two chicks of unknown sex) were used to constitute the vocalization databank used as stimuli (previously described in (29))**".

Overall, the Nature Communications format (methods after the results) seems poorly suited for this particular manuscript because to understand the results one needs to know the methods, to understand the methods one needs to read the supplement – and after 54 pages manuscript text it is still not clear to me exactly what was tested and how. Tables with stimulus information, and visualisation of the procedure (for example flow charts would help), as is the methods provide not enough information for replication and the manuscript will be difficult to read by a wider audience.

To improve the description of the stimulus construction, we added 2 columns to the Supplementary methods table 1 that indicate the sum of the silence period per stimulus on average for each type of behavioral test and the number of vocalizations in a given stimulus for each type of test. We added a description of how the inter-call intervals were calculated for each stimulus in the Acoustic stimuli section of the supplemental methods: "**The 5 or 2 intervals between renditions in a given stimulus were randomly drawn from a uniform distribution.**"

We added Figure 2 so that the typical sequence of behavioral tests performed by bird subjects is better described.

For replication purposes, the matlab code used to construct stimuli and run experiments is made freely available upon request to the authors. This is now clearly indicated in the method section line 229:

To investigate the ability of adult zebra finches to discriminate vocalizers based on acoustic cues, we used a forced-choice operant conditioning paradigm previously developed in our laboratory and described elsewhere (see (11, 22), Fig. 1 and Movie 1, **custom Matlab code available upon request to the authors**).

Butler NE et al. 2017: Lack of alarm calls in a gregarious bird: models and videos of predators prompt alarm responses but no alarm calls by zebra finches. *Behav Ecol Sociobiol* 71: Unsp 113.

10.1007/s00265-017-2343-z

Fernandez MSA et al. 2016: A New Semi-automated Method for Assessing Avian Acoustic Networks Reveals that Juvenile and Adult Zebra Finches Have Separate Calling Networks. *Frontiers in Psychology* 7: 1816. 10.3389/fpsyg.2016.01816

Kniel N et al. 2016: Sex-Specific Audience Effect in the Context of Mate Choice in Zebra Finches. *Plos One* 11: e0147130. 10.1371/journal.pone.0147130

Ma SW et al. 2017: Power-law scaling of calling dynamics in zebra finches. *Scientific Reports* 7: 8397. 10.1038/s41598-017-08389-w

Perez EC et al. 2016: Corticosterone triggers high-pitched nestlings' begging calls and affects parental behavior in the wild zebra finch. *Behav Ecol* 27: 1665-1675. 10.1093/beheco/arw069

References :

1. Elie JE & Theunissen FE (2016) The vocal repertoire of the domesticated zebra finch: a data-driven approach to decipher the information-bearing acoustic features of communication signals. *Anim Cogn* 19(2):285-315.
2. Cynx J & Nottebohm F (1992) Role of gender, season, and familiarity in discrimination of conspecific song by zebra finches (*Taeniopygia guttata*). *Proceedings of the National Academy of Sciences* 89(4):1368-1371.
3. Clayton NS (1988) Song discrimination learning in zebra finches. *Animal Behaviour* 36(4):1016-1024.
4. Miller DB (1979) Long-term recognition of father's song by female zebra finches. *Nature* 280(5721):389-391.
5. Miller DB (1979) The acoustic basis of mate recognition by female Zebra finches (*Taeniopygia guttata*). *Animal Behaviour* 27(Part 2):376-380.
6. Vignal C, Mathevon N, & Mottin S (2004) Audience drives male songbird response to partner's voice. *Nature* 430(6998):448-451.
7. Vignal C, Mathevon N, & Mottin S (2008) Mate recognition by female zebra finch: Analysis of individuality in male call and first investigations on female decoding process. *Behav Process* 77(2):191-198.
8. Vicario DS, Naqvi NH, & Raksin JN (2001) Sex differences in discrimination of vocal communication signals in a songbird. *Animal Behaviour* 61(4):805-817.
9. Forstmeier W, Burger C, Temnow K, & Deregnacourt S (2009) The genetic basis of zebra finch vocalizations. *Evolution* 63(8):2114-2130.
10. Mouterde SC, Theunissen FE, Elie JE, Vignal C, & Mathevon N (2014) Acoustic Communication and Sound Degradation: How Do the Individual Signatures of Male and Female Zebra Finch Calls Transmit over Distance? *Plos One* 9(7).

11. Mouterde SC, Elie JE, Theunissen FE, & Mathevon N (2014) Learning to cope with degraded sounds: female zebra finches can improve their expertise in discriminating between male voices at long distances. *The Journal of Experimental Biology* 217(17):3169-3177.
12. Zann RA (1996) *The Zebra Finch : A Synthesis of Field and Laboratory Studies* (Oxford University Press, Oxford).
13. Mulard H, Vignal C, Pelletier L, Blanc A, & Mathevon N (2010) From preferential response to parental calls to sex-specific response to conspecific calls in juvenile zebra finches. *Animal Behaviour* 80(2):189-195.
14. Jacot A, Reers H, & Forstmeier W (2010) Individual recognition and potential recognition errors in parent-offspring communication. *Behavioral Ecology and Sociobiology* 64(10):1515-1525.
15. Reers H, Jacot A, & Forstmeier W (2011) Do zebra finch parents fail to recognise their own offspring? *Plos One* 6(4):e18466.
16. Levrero F, Durand L, Vignal C, Blanc A, & Mathevon N (2009) Begging calls support offspring individual identity and recognition by zebra finch parents. *Cr Biol* 332(6):579-589.
17. Ligout S, Dentressangle F, Mathevon N, & Vignal C (2016) Not for Parents Only: Begging Calls Allow Nest-Mate Discrimination in Juvenile Zebra Finches. *Ethology* 122(3):193-206.
18. Elie JE, *et al.* (2010) Vocal communication at the nest between mates in wild zebra finches: a private vocal duet? *Animal Behaviour* 80(4):597-605.
19. D'Amelio PB, Trost L, & ter Maat A (2017) Vocal exchanges during pair formation and maintenance in the zebra finch (*Taeniopygia guttata*). *Frontiers in Zoology* 14(1):13.
20. Gill LF, Goymann W, Maat A, & Gahr M (2015) Patterns of call communication between group-housed zebra finches change during the breeding cycle. *Elife* 4.
21. D'Amelio PB, Klumb M, Adreani MN, Gahr ML, & ter Maat A (2017) Individual recognition of opposite sex vocalizations in the zebra finch. *Sci Rep-Uk* 7.
22. Perez EC, *et al.* (2015) Physiological resonance between mates through calls as possible evidence of empathic processes in songbirds. *Horm Behav* 75:130-141.

REVIEWERS' COMMENTS:

Reviewer #1 (Remarks to the Author):

The authors have made good efforts to adapt their manuscript to the reviewers' requests. I am broadly happy with the manuscript now.

I have one reservation: I argued that the birds were not demonstrated to use "names", and the authors changed this to "labels". It seems to me to be also odd to use the term "labels". The analogy with human laughter given by the authors themselves is illustrative: if you laugh and I recognise you from that sound, is that a label? Not in any conventional sense. It is a signature. "Signature" is a well-understood term in this discipline.

The authors argue in their reply to Reviewer 3 that the signatures are shown to be embedded using "active control". This is true in the sense that motor control is involved, but the paper does not show that an individual bird is able behaviourally to modify the signature in any one particular call (whether volitionally or otherwise).

I leave it to the editor to judge how strongly to treat this reservation.

Reviewer #3 (Remarks to the Author):

The authors have improved the manuscript substantially but there still are some passages and some missing methodological detail – addressing these points will greatly increase further accessibility of the study and data (the manuscript remains a difficult read).

Issues in the reply and general points

Re referee 1:

- Effect size 0.2 is small in biological sciences – especially when as here experimental (not epidemiological) data are concerned.**
- Terminology label like/name: the referee is spot-on here, and it is good the authors removed 'name' but label-like is still ambiguous (there is an individual signature in the sounds but not a label, which, like a name, is by definition arbitrary)**

I still think it worth to discuss the question as to whether it is a biologically sound expectation to assume that all vocalizations of an individual bear the same individual signature: Human spoken sounds are individually recognisable, but do not predict singing voices or laughter (Lavan et al. 2016, Lavan et al. 2018) – and whispering remains difficult to recognise individually (likewise screams of pain or fear – upon first exposure- cannot necessarily be recognised as belonging to a particular speaker. From a theoretical point of view, individual identity is likely to be selected for in some contexts (e.g. territory defence) but not necessarily in others (alarm calling) but also in deceptive contexts (see e.g. Dalziel & Magrath 2012)

33 work in bats showing voice recognition in more than one vocalisation seems relevant here? e.g. (Prat et al. 2016) or across large song type repertoires in songbirds? (Moser - Purdy & Mennill 2016)

49 ID cues in laughing bouts are not under voluntary control?

57 communicating 'state' is not the sole function of these calls – some calls have different functions

99 this prediction cannot be understood by a reader – mention somewhere before what method you are using and what interruption behaviour is..

108 Re and NoRe not previously defined

136PAF and LDA not previously defined

148/9 and also 156ff leaving discussing and interpreting the results to the discussion, remove here

160 is an alternative that they learned each call separately and then formed a metacategory?

178 if the lowest frequencies (see above) is 1 kHz there will be harmonics at 2 kHz – how can they not be modified if the 3 kHz can?

189 is this an ecological or social pressure? What evidence do we have for this? Conformity (flock members) could also be a selection pressure? Moreover, a part from the distance calls, looking/listening to other zebra finch calls one does not get the impression that there was strong selection pressure on individuality, the comparison with dolphin signature whistles (that are very individually distinct and not recognisable as a generic call type) is odd here.

219ff if there was selection to learn to sound individually different than why only in one sex and not the other?

234 too little info on subjects (see for standards reporting on experimental animals see Kilkenny et al. 2010) what was the average age (male/females) of tested subjects and were all birds of the same origin (same breeding colony?) – this is important to mention as this might influence (dis)similarity of calls

243 If there were 3 or 6 calls: what rule determined when there were 3 and when 6 ? And is this within or between call type/subject variation? and why were there these 2 categories? Why a 6s window? Even if playing back 6 calls most of the calls are so short that most of the 6 s window would have been silence? Is it possible that a bird pecked and waited more than 5 s to hear a sound as the 3-6 calls happened to be at the end of the 6 s? Regardless of the info in the supplement please also indicate here how many stimuli there were per individual vocalizer used during training (e.g. while learning to discriminate the Re vs. NoRe how many different stimuli were they hearing?)

252 'refraining from interrupting' if there were 3-6 calls (most <50ms) in 6s window – how can a bird assess if it is interrupting? Pauses within the stimulus presentation will be several seconds long?

254/55 formula not self-explanatory: you state that calculating the odds ratio involves dividing the other two odds ratios: Please state how these were calculated (i.e. which numbers went into the calculation steps – total pecks per stimulus, or per stimulus category? per session? average or totals over all sessions?), likewise give the formula for

calculating the probabilities.

257 please indicate the fasting time also in day time hours as zebra finch feeding activity varies circadiannally – most importantly was this 15 daylight hours or was this period including night hours (lights off?)

Fig 1B: this is a graph and not an ethogram?

477 Unclear, does this mean that there was a new shaping session (grey square) with 2 songs before each test series with calls even if the birds has learned the task in a previous series?

541 p-value: reduce number of decimals and use more standard format? (value is potentiated to `– 186'?) 543 dito – reduce decimals for p-value

References

Dalziell AH and Magrath RD 2012: Fooling the experts: accurate vocal mimicry in the song of the superb lyrebird, *Menura novaehollandiae*. *Anim Behav* 83: 1401-1410.

10.1016/j.anbehav.2012.03.009

Kilkenny C et al. 2010: Improving bioscience research reporting: The ARRIVE guidelines for reporting animal research. *PLoS Biol* 8: e1000412.

Lavan N et al. 2016: Impaired Generalization of Speaker Identity in the Perception of Familiar and Unfamiliar Voices. *Journal of Experimental Psychology-General* 145: 1604-1614. 10.1037/xge0000223

Lavan N et al. 2018: Impoverished encoding of speaker identity in spontaneous laughter. *Evolution and Human Behavior* 39: 139-145. 10.1016/j.evolhumbehav.2017.11.002

Moser-Purdy C and Mennill DJ 2016: Large vocal repertoires do not constrain the dear enemy effect: a playback experiment and comparative study of songbirds. *Anim Behav* 118: 55-64. 10.1016/j.anbehav.2016.05.011

Prat Y et al. 2016: Everyday bat vocalizations contain information about emitter, addressee, context, and behavior. *Scientific Reports* 6: 39419. 10.1038/srep39419

Dear reviewers,

Thank you for your second round of comments and your hard work. We have made additional changes following your suggestions and requests (as well as those of the Editor).

Reviewer #1 (Remarks to the Author):

The authors have made good efforts to adapt their manuscript to the reviewers' requests. I am broadly happy with the manuscript now.

I have one reservation: I argued that the birds were not demonstrated to use "names", and the authors changed this to "labels". It seems to me to be also odd to use the term "labels". The analogy with human laughter given by the authors themselves is illustrative: if you laugh and I recognise you from that sound, is that a label? Not in any conventional sense. It is a signature. "Signature" is a well-understood term in this discipline.

We have changed "label" to "signature" throughout the text.

The authors argue in their reply to Reviewer 3 that the signatures are shown to be embedded using "active control". This is true in the sense that motor control is involved, but the paper does not show that an individual bird is able behaviourally to modify the signature in any one particular call (whether volitionally or otherwise).

We agree with this comment. We had used the term "active control" to contrast it with passive filtering that occurs as the result of individual specific morphology. Since "active" has connotations of "plasticity", we are now just saying "neural control" with the understanding that this neural control could be passive (genetic or simply correlated with the motor program for each call type) or active (volitional).

I leave it to the editor to judge how strongly to treat this reservation.

Reviewer #3 (Remarks to the Author):

The authors have improved the manuscript substantially but there still are some passages and some missing methodological detail – addressing these points will greatly increase further accessibility of the study and data (the manuscript remains a difficult read).

We have further modified the results section to address this issue. The results now include a first section that succinctly describes the experimental design. We also brought back in the main manuscript one of the key supplementary figure to facilitate the reading flow.

Issues in the reply and general points

Re referee 1:

- Effect size 0.2 is small in biological sciences – especially when as here experimental (not epidemiological) data are concerned.

We agree that conventions on effect sizes are just that. It is up to the field and individual readers to decide whether an effect size is large or small. We note that we are talking here of a correlation involving behavioral performance obtained by averaging all trials (and not for example, whether birds can or cannot discriminate above chance or using times of peak performance). It will be interesting to compare effect sizes measured by other groups in similar behavioral tasks. In the paper, without explicitly commenting on the size of the effect, we give possible reasons for why the correlation could be only 0.2. We believe that it is mostly due to intrinsic behavioral variability in this task..

- Terminology label like/name: the referee is spot-on here, and it is good the authors removed 'name' but label-like is still ambiguous (there is an individual signature in the sounds but not a label, which, like a name, is by definition arbitrary)

As indicated above we decided to change "label" to "signature" throughout the text such as not to mislead the reader on terminology issues.

I still think it worth to discuss the question as to whether it is a biologically sound expectation to assume that all vocalizations of an individual bear the same individual signature: Human spoken sounds are individually recognisable, but do not predict singing voices or laughter (Lavan et al. 2016, Lavan et al. 2018) – and whispering remains difficult to recognise individually (likewise screams of pain or fear – upon first exposure – cannot necessarily be recognised as belonging to a particular speaker. From a theoretical point of view, individual identity is likely to be selected for in some contexts (e.g. territory defence) but not necessarily in others (alarm calling) but also in deceptive contexts (see e.g. Dalziell & Magrath 2012)

Our intent in this study was to explore the question of vocal recognition throughout a repertoire without any a priori as to whether it would be found for all call types. Indeed, the coding of individual identity is not necessarily under evolutive pressure for all vocalizations and it is not expected that all vocalizations from an individual bear an individual signature. However, given our experimental finding that the zebra finch does discriminate vocalizers irrespective of call type, we were then surprised to find that the most parsimonious mechanism to encode identity in all vocalizations (passive filtering cues), that had also been suggested by a previous study in cervids (Reby et al 2006), was not the mechanism at stake in zebra finches. We thank the reviewer for pointing us to the studies by Lavan which are very relevant to our manuscript. We included both of the suggested papers as references as well as another manuscript from the same author.

33 work in bats showing voice recognition in more than one vocalisation seems relevant here? e.g. (Prat et al. 2016) or across large song type repertoires in songbirds? (Moser-Purdy & Mennill 2016)
We added the bat reference as an example of investigation of the identity coding potential of vocalizations. Note that this study is based solely on acoustical analysis and that it does not show any behavioral evidence of vocal recognition. Regarding the vocal recognition of songbird across a large repertoire of songs, we are already citing Briefer et al 2008, which performed a similar demonstration of the dear-enemy effect in Skylarks. Because it's an earlier study that is also cited by Moser-Purdy & Mennill 2016 we prefer to keep that citation as an example of vocal recognition across a large repertoire of songs in songbirds.

49 ID cues in laughing bouts are not under voluntary control?

We cited examples of individual signatures that are unique to specific call-types including ID cues in laughing bouts in humans. We are not making any specific statements about these being under voluntary control. Laughing bouts in humans can both be spontaneous and voluntary and the ID signatures appear to be different for these two contexts. We have slightly reworded that sentence to be more precise.

57 communicating 'state' is not the sole function of these calls – some calls have different functions

We agree with the reviewer that this description was too succinct: communicating state is only one of the functions of vocal communication. We changed it to "states, intents or needs".

99 this prediction cannot be understood by a reader – mention somewhere before what method you are using and what interruption behaviour is..

Agreed. We added a paragraph on the experimental design at the beginning of the result section that complement figure 1 and 2, and that clearly explains the different tests that the bird are performing and what interruption behavior is. The reader will now understand the main text without having to read the figure legends.

108 Re and NoRe not previously defined

Re and NoRe are now well defined both in the legend of figures and in the experimental design section.

136PAF and LDA not previously defined

The same paragraph on experimental design at the beginning of the results section now defines these terms.

148/9 and also 156ff leaving discussing and interpreting the results to the discussion, remove here

We agree that the statement that was on line 148/9 would usually belong to the discussion section, but we prefer to leave it here as it is not central to the ideas that are addressed in the discussion section. As for the section started on line 156, we kept the sentences that summarize the results and moved the sentences that discuss the potential utility of recognition across multiple call type in the discussion.

160 is an alternative that they learned each call separately and then formed a metacategory?

Yes! This is exactly what we are proposing.

178 if the lowest frequencies (see above) is 1 kHz there will be harmonics at 2 kHz – how can they not be modified if the 3 kHz can?

We are not sure if we completely understand this comment. We are here talking about the resonant frequencies of the filter not of the source. The source generates a fundamental with harmonics. Resonant peaks in the filter are not necessarily harmonically related as they are created by different “sections” of the upper vocal tract. Variations across individual in vocal tract length correspond to shifts in the resonant peaks in the zebra finch between 5kHz and 6kHz (1kHz centered around 5.5kHz). In addition, zebra finches can actively control their upper vocal tract by modifying the volume of OEC and changing the gape of their beak. Anatomical experiments and modeling have shown that these modifications can generate filter resonances between 3 and 10 kHz (Riede et al, 2013). Finally acoustic analyses of the call-types have shown that call-types are characterized and distinguished by different or idiosyncratic resonance peaks in the frequency range of 500Hz to 8kHz measured (Elie and Theunissen, 2016). If animals were to rely on the resonances in the 5 to 6kHz to determine an individual identity (the only frequency range that is affected by the size of the animal according to the variations of their vocal tract length), then they would have to take into account the contamination generated by the peaks that characterize the call-type.

189 is this an ecological or social pressure? What evidence do we have for this? Conformity (flock members) could also be a selection pressure? Moreover, a part from the distance calls, looking/listening to other zebra finch calls one does not get the impression that there was strong selection pressure on individuality, the comparison with dolphin signature whistles (that are very individually distinct and not recognisable as a generic call type) is odd here.

Thank you for requiring more precision on this discussion topic. We changed “ecological pressure” to “socio-ecological pressure” as we meant that individual recognition for all call types could have evolved because it gives advantages for forming specific social bonds (e.g. mate selection and recognition, neighbour recognition or young recognition). Beyond this social advantages, ecological pressures such as the need to find each other in noisy environments out of visual contact could also play a role in shaping the ID signature. We are clearly not proving these evolutionary hypothesis in this manuscript but suggest it as a possibility. This discussion naturally comes out from our discovery based on acoustic analysis and behavioral experiments, the calls from the zebra finch are different from one individual to the other, irrespective of the call-type. Note that other work has discussed both potential ecological pressures for shaping the individual signature of bird calls (eg. Mathevon et al. 2008, Mouterde et al, 2014). In zebra finches this is true for DC (Mouterde et al, 2014) but it is also reflected in the use of tet and stack calls during mate behavior (D’Amelio et al. 2017). Also the social pressure for sounding similar is not necessarily exclusive of pressure for conformity and we are not make such claim. For example, songbirds can both produce learned vocalizations (song) that have group signatures (dialects) and individual signatures.

The reference to the dolphin whistles is relevant for a more specific point that we are making on potential evolutionary forces: Dolphins and Zebra finches do not rely on passive spectral filtering for the coding of identity. We propose that this might be the case because of ecological (living under-water for dolphins) and morphological (being very small for zebra finches) constraints that make the passive filtering an inefficient or unreliable signal for the coding of identity.

219ff if there was selection to learn to sound individually different than why only in one sex and not the other?

This is a good question that is of a great interest to not only us but to the large community who is interested in the evolution of vocal imitation (and more specifically song learning and production) in male songbirds in temperate regions. We don’t know! Clearly sexual selection could have driven the evolution of the learned complex song. A learned song can also provide better identification that is also beneficial (for flock recognition and/or individual recognition) and this could have played a role in the evolution of vocal imitation. Territorial singing is clearly directed (also) to other males and individual recognition is very important in these situations. Once a neural system is in place for song learning, it could be used to further shape the individual Distance Call in males. We also believe that vocal plasticity and imitation in females for other calls should be examined more closely. It is well known that both males and female songbirds have song system nuclei but that these can be greatly reduced in females (Fortune et al., 2011). The origin of vocal learning is still a mystery but once it has evolved it could be advantageous for many aspects of animal social and sexual behaviors.

234 too little info on subjects (see for standards reporting on experimental animals see Kilkenney et al. 2010) what was the average age (male/females) of tested subjects and were all birds of the same origin (same breeding colony?) – this is important to mention as this might influence (dis)similarity of calls

We added more information on both the subjects and the vocalizers of our study at time of audio-recordings, including their age, sex and origin.

243 If there were 3 or 6 calls: what rule determined when there were 3 and when 6? And is this within or between call type/subject variation? and why were there these 2 categories?

The rules depended on the natural patterns of production and is well described lines 47 to 50 of the supplemental method.

Why a 6s window?

The length of the window was optimized for this conditioning task (birds are able to learn this task quickly and perform very well with easy discriminations). We believe that this duration works well in this behavioral paradigm in the sense of preventing false positives. With shorter trial times, birds would “refrain from interrupting” by their natural pauses. Longer trial times could presumably frustrate the animal and make the task too difficult (e.g. in terms of working memory). When designing our task we varied this window length and obtained good results at 6s.

Even if playing back 6 calls most of the calls are so short that most of the 6 s window would have been silence?

Is it possible that a bird pecked and waited more than 5 s to hear a sound as the 3-6 calls happened to be at the end of the 6 s?

Each trial always started with a call rendition and ended with a call rendition (or call bout). The other 4 calls occur at random times (uniform distribution) between the start and end times (with no overlapping). We reworded the methods section to more clearly describe how the sounds are presented. The key element here is that the bird immediately hears a new call when he or she pecks.

Regardless of the info in the supplement please also indicate here how many stimuli there were per individual vocalizer used during training (e.g. while learning to discriminate the Re vs. NoRe how many different stimuli were they hearing?).

The section on experimental design now added at the beginning of the results described that 2 song renditions each from 2 different birds were used to first train the birds to learn how to use the operant system to get rewards.

252 ‘refraining from interrupting’ if there were 3-6 calls (most <50ms) in 6s window – how can a bird assess if it is interrupting? Pauses within the stimulus presentation will be several seconds long?

As mentioned above, we now more explicitly describe in the supplementary methods (line 50) how sounds are played back during the trials. Importantly, as soon as a bird interrupted a stimulus, it could hear the first rendition of the just-triggered stimulus.

254/55 formula not self-explanatory: you state that calculating the odds ratio involves dividing the other two odds ratios: Please state how these were calculated (i.e. which numbers went into the calculation steps – total pecks per stimulus, or per stimulus category? per session? average or totals over all sessions?), likewise give the formula for calculating the probabilities.

We now indicate how the probability of interruption is calculated (line 303 in the Methods):

The probabilities of interruption were calculated by dividing the number of interrupted stimuli of a given type by the total number of triggered stimuli of that type.

The odds ratio was calculated as a time running value and as an average for the 3 daily sessions.

257 please indicate the fasting time also in day time hours as zebra finch feeding activity varies circadianly – most importantly was this 15 daylight hours or was this period including night hours (lights off?)

We added now this information in the methods of the main manuscript line 306-311.

Fig 1B: this is a graph and not an ethogram?

We replaced ethogram by schematic

477 Unclear, does this mean that there was a new shaping session (grey square) with 2 songs before each test series with calls even if the birds has learned the task in a previous series?

Yes we did repeat the shaping so that the experience before each session was identical. Note that order of sessions such a male vs female vocalizers was randomized. As the reviewer pointed it out this is shown on figure 2 (yes it is correct) and it is also described in the supplementary methods line 69 to 72.

541 p-value: reduce number of decimals and use more standard format? (value is potentiated to '– 186'?)
543 dito – reduce decimals for p-value

Exact p-values are now often requested by publishers. Although these are nonsensical from a statistical point of view, they do provide a trace that can be used for reproducibility (akin to hash codes). We are happy to change this and leave this decision to the editor.

References

Dalziel AH and Magrath RD 2012: Fooling the experts: accurate vocal mimicry in the song of the superb lyrebird, *Menura novaehollandiae*. *Anim Behav* 83: 1401-1410. 10.1016/j.anbehav.2012.03.009

Kilkenny C et al. 2010: Improving bioscience research reporting: The ARRIVE guidelines for reporting animal research. *PLoS Biol* 8: e1000412.

Lavan N et al. 2016: Impaired Generalization of Speaker Identity in the Perception of Familiar and Unfamiliar Voices. *Journal of Experimental Psychology-General* 145: 1604-1614. 10.1037/xge0000223

Lavan N et al. 2018: Impoverished encoding of speaker identity in spontaneous laughter. *Evolution and Human Behavior* 39: 139-145. 10.1016/j.evolhumbehav.2017.11.002

Moser-Purdy C and Mennill DJ 2016: Large vocal repertoires do not constrain the dear enemy effect: a playback experiment and comparative study of songbirds. *Anim Behav* 118: 55-64. 10.1016/j.anbehav.2016.05.011

Prat Y et al. 2016: Everyday bat vocalizations contain information about emitter, addressee, context, and behavior. *Scientific Reports* 6: 39419. 10.1038/srep39419

References from Reply:

Riede, T., N. Schilling and F. Goller (2013). "The acoustic effect of vocal tract adjustments in zebra finches." *Journal of Comparative Physiology a-Neuroethology Sensory Neural and Behavioral Physiology* **199**(1): 57-69.

Elie, J. E. and F. E. Theunissen (2016). "The vocal repertoire of the domesticated zebra finch: a data-driven approach to decipher the information-bearing acoustic features of communication signals." *Animal Cognition* **19**(2): 285-315.

Mouterde, S. C., F. E. Theunissen, J. E. Elie, C. Vignal and N. Mathevon (2014). "Acoustic communication and sound degradation: how do the individual signatures of male and female zebra finch calls transmit over distance?" *PLoS One* **9**(7): e102842.

Mathevon, N., T. Aubin, J. Vielliard, M.-L. da Silva, F. Sebe and D. Boscolo (2008). "Singing in the Rain Forest: How a Tropical Bird Song Transfers Information." *PLoS One* **3**(2): Article No.: e1580.

D'Amelio, P. B., M. Klumb, M. N. Adreani, M. L. Gahr and A. ter Maat (2017). "Individual recognition of opposite sex vocalizations in the zebra finch." *Scientific Reports* **7**.

Fortune, E. S., C. Rodriguez, D. Li, G. F. Ball and M. J. Coleman (2011). "Neural Mechanisms for the Coordination of Duet Singing in Wrens." *Science* **334**(6056): 666-670.